# *Lacticaseibacillus rhamnosus* MS27 Potentially Prevents Ulcerative Colitis Through Modulation of Gut Microbiota

**DOI:** 10.3390/ijms262311397

**Published:** 2025-11-25

**Authors:** Jie Zhang, Jiakun Shen, Linbao Ji, Peng Tan, Chunchen Liu, Xiujun Zhang, Xi Ma

**Affiliations:** 1School of Public Health, North China University of Science and Technology, Tangshan 063210, China; 71819@bvca.edu.cn (J.Z.); liuchunchenlcc@163.com (C.L.); 2State Key Laboratory of Animal Nutrition, College of Animal Science and Technology, China Agricultural University, Beijing 100193, China; shenjiakun2008@126.com (J.S.); jilinbao0126@cau.edu.cn (L.J.); tanpeng1995@cau.edu.cn (P.T.)

**Keywords:** *Lacticaseibacillus rhamnosus*, probiotics, ulcerative colitis, microbiota, transcriptomic profiles, immune responses

## Abstract

(1) This study explored *Lacticaseibacillus rhamnosus* MS27, a newly isolated strain, as a potential probiotic candidate for alleviating the onset and severity of ulcerative colitis (UC). (2) *L. rhamnosus* MS27 was isolated and subjected to biochemical identification, antibiotic sensitivity testing, and antibacterial activity assessment. Dextran sulfate sodium (DSS) colitis model mice were used to evaluate its alleviating effects. In this study, 16S rRNA microbiome and eukaryotes reference transcriptome analyses were conducted to investigate its impact on intestinal microbial ecology and potential molecular mechanisms. (3) *L. rhamnosus* MS27 exhibits high acid tolerance at pH 3.23 and maintains a high viable bacterial count for 24 h. It can utilize sucrose, lactose, maltose, inulin, esculin, salicin, and mannitol but not raffinose, and it is sensitive to carbenicillin, erythromycin, tetracycline, chloramphenicol, clindamycin, and penicillin. It effectively increases the abundance of beneficial microbes, particularly *Akkermansia*, *Muribaculaceae,* and *Limosilactobacillus reuteri* (*p* < 0.05), while significantly reducing microorganisms linked to human pathogens causing diarrhea and gastroenteritis (*p* < 0.05). Transcriptomic analysis demonstrated that the expression levels of *Igkv16-104* and *C1qtnf3* were significantly downregulated in the presence of *L. rhamnosus* MS27 treatment compared to DSS treatment alone (*p* < 0.05). Further analysis revealed significant differences in genes related to immune functions, antigen presentation, and immune cell markers, indicating potential protein–protein interaction networks, particularly among genes of the major histocompatibility complex (MHC). (4) *L. rhamnosus* MS27, as a novel strain, demonstrates a significant capacity to alleviate inflammatory phenotypes. *L. rhamnosus* MS27 exhibits distinctive metabolic characteristics in lactic acid utilization, acetic acid and oleic acid production. Furthermore, it contributes to systemic homeostasis regulation by modulating *Turicibacter* to link intestinal microbiota composition with host immune function.

## 1. Introduction

Ulcerative colitis (UC) is a complex and multifactorial disease characterized by the interaction of genetic, environmental, and immunological factors. Its pathogenesis is associated with the dysbiosis of gut microbiota, leading to increased intestinal permeability and chronic inflammation, for which non-pathogenic enteric bacteria play a primary role in the phenotype of the UC and the severity of intestinal inflammation. Therefore, UC appears to arise from the disruption of the homeostatic equilibrium between the host’s mucosal immune system and the enteric microbiota, leading to an inappropriate immune response against non-pathogenic commensal bacteria [1,2]. During the various treatment processes for UC, conventional treatments, including 5-aminosalicylic acid (5-ASA), glucocorticoids, and biological agents, have failed or come with significant side effects for a substantial proportion of patients with UC. 5-ASA intolerance is associated with a risk of adverse clinical outcomes and intestinal microbial ecology dysbiosis in patients with UC; its common clinical adverse events include diarrhea, fever and rash [3], and more serious events such as renal toxicity, liver dysfunction, pancreatitis, pericarditis, pneumonia, severe skin reactions, etc. [4]. Glucocorticoid can induce osteoporosis (GIOP) in patients [5]; biological agents such as monoclonal antibodies infliximab, which work as antitumor necrosis factor (TNF) agents, may increase the risk of infection in patients and the risk of certain malignant tumors (such as lymphoma); and responses to infliximab therapy are highly variable among individuals, too. Formation of antidrug antibodies (ADAs) due to the failure of anti-TNF therapy is another thorny issue, especially in children, as the available alternative treatment options are limited [6]. However, in patients with UC who have undergone ileal pouch-anal anastomosis but continue to experience intermittent symptoms of pouchitis, probiotics are recommended to prevent recurrence [7]. The role of probiotics in the prevention of UC is a meaningful concept that should be explored.

Meanwhile, the balance of gut microbiota is crucial in UC, owing to its function in modulating the immune system and maintaining intestinal homeostasis. Dysbiosis, or an imbalance in the gut microbiota, is a hallmark of UC, characterized by a decrease in beneficial bacteria and an increase in pathogenic bacteria. Therefore, there is a growing interest in exploring probiotics, such as *Lacticaseibacillus rhamnosus*, as a safer and more natural therapeutic option to help restore this balance [8].

*Lacticaseibacillus rhamnosus* has been shown to benefit human health, based on numerous studies, and its advantages include immune modulation, diarrhea prevention, allergy reduction, cholesterol levels regulation, and the ability to maintain the microbiota balance, etc. [9,10,11,12]. *Lacticaseibacillus rhamnosus* has also been extensively studied for its potential therapeutic effects across a range of health conditions. One of the most notable applications is in the treatment of gastrointestinal disorders, such as UC [13]. Different strains of *Lacticaseibacillus rhamnosus* may have distinct mechanisms of action in treating UC. For example, *Lacticaseibacillus rhamnosus* L34 can attenuate colitis severity through gut microbiota modulation and by decreasing the secretion of pro-inflammatory cytokines, while *Lacticaseibacillus rhamnosus* GG (LGG) suppresses colitis in a STING-dependent manner in Ly6C+ monocytes [14]. However, the specific mechanisms by which other strains exert their effects are still not fully understood.

In this study, we explored the role of *L. rhamnosus* MS27 d in alleviating inflammatory phenotypes and also explored the comprehensive characteristics of *L. rhamnosus* MS27 and its effectiveness of action via the perspective of microbiomics and untargeted transcriptomics. Through multi-omics association analysis and microbiota–immune axis elucidation, new research targets will be established for advancing more in-depth mechanism analysis on *L. rhamnosus* MS27.

## 2. Results

### 2.1. Strain Characteristics of L. rhamnosus MS27 as a New Probiotic Candidate

The morphology of *L. rhamnosus* MS27 colonies on MRS solid medium is a milky white circular colony with a diameter of 1 to 2 mm. Colonies are smooth and moist, slightly raised with a certain degree of stickiness (Figure 1A). Gram staining and microscopic examination proved that *L. rhamnosus* MS27 is rod-shaped, bluish-purple, non-spore-forming Gram-positive bacteria (Figure 1B), without fimbriae and flagella (Figure 1C). Initial DNA sequencing and blasting with the National Center for Biotechnology Information (NCBI) database proved that this strain belongs to the genus *Lacticaseibacillus,* with a similarity of 99.80%. The phylogenetic tree indicates that the MS27 strain should belong to the category of *Lacticaseibacillus rhamnosus*; however, it is a new strain with different genes from other *Lacticaseibacillus rhamnosus* strains (Figure 1D).

*L. rhamnosus* MS27 shows an initial increase in viable cell count followed by gradual stabilization as the growth time prolongs. The highest viable cell counts of approximately 1.47 × 10^9^ CFU/mL is reached at 22 h. The OD600 curve follows an S-shaped pattern. It can be inferred that the logarithmic growth period of MS27 is approximately 4 h to 10 h. The pH results indicate that MS27 is an acid-producing bacterium. As time progresses, the pH continuously decreases, and after 24 h of growth, the pH drops to 3.23. *L. rhamnosus* MS27 has high acid tolerance (Figure 1E), good growth characteristics (Figure 1F), and a high viable bacteria number (Figure 1G) within 24 h.

### 2.2. Utilization of Carbohydrates by L. rhamnosus MS27

A biochemical analysis was conducted to compare the carbohydrate metabolic characteristics of several *Lacticaseibacillus rhamnosus* strains. The results confirmed that MS27 has positive reactions to lactose, sucrose, maltose, inulin, synanthrin, esculin, salicin, mannitol, and sorbitol, and a negative reaction to raffinose. Moreover, the ability of MS27 to utilize lactose is different from other strains. The respective results are shown in Table 1.

### 2.3. Antibiotic Susceptibility and Antimicrobial Activity of L. rhamnosus MS27

The drug resistance of the strains was evaluated by selecting representative categories of antibiotics for drug sensitivity analysis, mainly as follows: penicillins (penicillin, carbobenzicillin), macrolides (erythromycin), tetracyclines (tetracycline), aminoalcohols (chloramphenicol), and lincomycin (clindamycin). Among them, a diameter of the inhibition zone of less than 15 mm was considered drug-resistant (R); a diameter between 16 mm and 20 mm was moderately sensitive (I); and a diameter greater than 20 mm was sensitive (S). The results shown in Figure 1H and Table 2 indicate that *L. rhamnosus* MS27 is sensitive to carbenicillin, erythromycin, and tetracycline, and is moderately sensitive to chloramphenicol, clindamycin, and penicillin. These results indicate that MS27 does not have obvious drug resistance and is a good probiotic.

To further explore the active ingredients that play a role in *L. rhamnosus* MS27, we test the effects of the bacterial solution (labeled as 1), the fermentation supernatant (labeled as 2), and the bacterial cells (labeled as 3) on the common pathogenic bacteria, which are *Staphylococcus aureus* ATCC 25923, *Escherichia coli* K88, and *Salmonella* SL1344. These results are shown in Figure 1I and indicate that *L. rhamnosus* MS27 has obvious inhibitory effects on common pathogenic bacteria, such as *Staphylococcus aureus*, *Escherichia coli,* and *Salmonella*. The inhibitory substances are produced and secreted by the bacteria into the fermentation supernatant, and the bacteria themselves have no obvious inhibitory effect.

### 2.4. Alleviation on the DSS-Induced Colitis of L. rhamnosus MS27

In order to explore whether the addition of *L. rhamnosus* MS27 has preventive effects on ulcerative colitis occurrence and development, the mice in the treatment group (LR+DSS) were intragastrically administered *L. rhamnosus* MS27 for 14 days, while the other two groups were all intragastrically administered normal saline at the same time to maintain the same stimulation conditions. During 7–14 d, the mice in the LR+DSS group and the DSS group drank water with 3% DSS (Figure 2A). The results showed that there was no significant difference in the body weight of mice among three groups during the entire experimental period. Although there was a certain degree of decline in mice body weight during the DSS treatment period, gavage *L. rhamnosus* MS27 did not affect body weight of mice significantly (*p* > 0.05) (Figure 2B). However, gavage *L. rhamnosus* MS27 significantly reduced the decline in weight gain ratio after DSS treatment, *p* < 0.05 (Figure 2C). In terms of liver index, there was no significant difference among the three groups (*p* > 0.05) (Figure 2D).

Analysis of colon length data demonstrated that oral administration of *L. rhamnosus* MS27 significantly attenuated the reduction in colon length induced by DSS treatment, *p* < 0.05 (Figure 2E), and helped to alleviate the severity of colitis. The results of histological sections also confirmed DSS treatment disrupted the original structural organization of the colon and led to a large amount of inflammatory cell infiltration, while gavage *L. rhamnosus* MS27 effectively alleviated the inflammatory phenotype of the colon tissue (Figure 2F).

### 2.5. Modulation on the Intestinal Microbial Ecology by L. rhamnosus MS27

*L. rhamnosus* MS27 improved the composition of colonic microflora and significantly increased microbial diversity via alpha diversity index group variance analysis, *p* = 0.01249 (Figure 3A). Carrying out hierarchical clustering analysis based on the beta diversity distance matrix showed that different groups have definite differences in community composition (Figure 3B) and that there is an extremely significant difference in samples in terms of inter-group community structure, *p* < 0.01 (Figure 3C). The raw reads of this sequencing are 2,775,649 and the valid data after quality control are 2,591,391, and the ratio is 93.36%.

The NST analysis of community structure in the sample comparative analysis shows that the preventive supplementation of *L. rhamnosus* MS27 significantly reduces the intestinal microbial dysbiosis caused by DSS (Figure 3D). Among the three groups, DSS induced colitis following the increase in *Faecalibaculum_rodentium*, *o_Clostridia*_UCG-014, and *g_Lachnospiraceae*_NK4A136_group, but reduced *Akkermansia_muciniphila, g_Lacticaseibacillus,* and *f_Muribaculaceae* (Figure 3E). However, *L. rhamnosus* MS27 effectively influenced these microbial abundance changes and adjusted them to have a similar change as the control group, especially for *Akkermansia, Muribaculaceae,* and *Limosilactobacillus reuteri* (Figure 3E,F). Further analysis of the component differences between the LR+DSS group and the DSS group indicated that *Lacticaseibacillus rhamnosus* increased significantly in the LR+DSS group, although its abundance value is not too high (Figure 3G). These differentially expressed microorganisms may represent key species that respond to environmental changes, and thus can be used to find the microbial biomarkers of the treatment group, helping to construct a prediction model.

Following the FAPROTAX functional annotation analysis of microbial communities, we proved that the intestinal microbial dysbiosis caused by DSS were comparative with human_pathogens_diarrhea and human_pathogens_gastroenteritis (Figure 3H). Further analysis of the differences among functional groups proved that *L. rhamnosus* MS27 induced reduction in the microorganisms of human_pathogens_diarrhea and human_pathogens_gastroenteritis is remarkable, *p* < 0.05 (Figure 3I).

### 2.6. Eukaryotes Reference Transcriptome Analysis of L. rhamnosus MS27

The transcriptome analysis indicated that the number of unique genes/transcripts in the gene set formed by adding *L. rhamnosus* MS27 was 543, accounting for 16.65%. Compared with the unique gene/transcripts formed by DSS treatment, the expression level of these unique gene/transcripts decreased by 22.48% (Figure 4A). Further analysis of gene/transcripts expression levels and their significance revealed that the administration of *L. rhamnosus* MS27 reduced the number of genes/transcripts upregulated by DSS treatment alone, decreasing from 1933 to 1548, and partially restored the expression of genes/transcripts downregulated by DSS exposure, increasing from 437 to 785. Analysis of the top 10 differentially expressed genes compared with the control group, *Cxcl9*, *Caspase12*, which are genes associated with IBD, increased with DSS treatment but decreased after the *L. rhamnosus* MS27 addition, but *Cxcl1* increased with the *L. rhamnosus* MS27 treatment (Figure 4B). Using the DSS group as the control, a comparison was made solely based on the effect of adding *L. rhamnosus* MS27 and the results showed that *Igkv16-104* and *C1qtnf3* were significantly downregulated (Figure 4C). Analyzing the expression levels of interleukin 6 (*IL 6*) found that the preventive supplementation of *L. rhamnosus* MS27 partially alleviates the ascendance of inflammatory indicators caused by DSS treatment (Figure 4D) and this result was also confirmed by quantitative real-time polymerase chain reaction (qPCR) (Figure 4E), although the alleviation effect is not significant enough (*p* > 0.05). Combined analysis of mucin gene expression indicated that pretreatment with *L. rhamnosus* MS27 prior to DSS-induced injury attenuated the downregulation of mucin 20 (*Muc20*) expression (Figure 4F), whereas mucin 2 (*Muc2*) expression was significantly reduced (*p* < 0.05) (Figure 4G).

In order to detect the differences in gene/transcript expression among different groups (samples), multiple group differential analysis was conducted and screened out dozens of genes significant to the DSS group and LR+DSS group. To further examine the correlations of these genes from the immune-related gene groups (involving inflammation, antigen presentation, immune cell markers, etc.), we used the STRING database to analyze these genes and generate a protein interaction (PPI, protein–protein interaction) network and the default value is ≥0.7 to reduce false positives. We used the MCL clustering algorithm of STRING to automatically group highly interconnected genes (Figure 4H). Compared with the samples randomly composed of proteins following the same size and distribution pattern, it is shown that there were significantly more interactions among proteins represented by these genes than expected. Among them, 20 genes of proteins are at least partially interrelated and have certain biological connections as four groups. They are *H2Ab1*, *H2Eb1*, *H2k1* (Figure 4I); *C1qa*, *C1qb*, *C1qc*, *C3ar1*, *C2* (Figure 4J); *Xcl1*, *Cx3cr1*, *Cxcl12*, *Ccl22*, *Cxcl16* (Figure 4K); and *Cd68*, *Mrc1*, *Csf1r*, *Tyrobp*, and *Fcer1g* (Figure 4L). Significance is indicated by FDR values of less than 0.05. The result of multiple group differential analysis shows that *L. rhamnosus* MS27 should have positive effects on immune regulation ability. This might be the underlying reason for its improvement in the inflammatory phenotype.

### 2.7. Effects on Immune Phenotypes of L. rhamnosus MS27

Whether *L. rhamnosus* MS27 could influence immune phenotypes after RNA-seq identified differentiated expressed genes and STING pathway analysis revealed their enrichment within the immune axis is the next pressing question. Thus, flow cytometry was performed to further explore its immunological phenotypes, which were predicted by bioinformatics. The results indicated that the population of splenic CD45^+^ CD11b^+^ MHC II^+^ cells increased significantly (*p* = 0.0382) compared with the control group following *L. rhamnosus* MS27 treatment (Figure 5A–D), suggesting a probable enhanced antigen-presenting capacity derived from the myeloid lineage. However, the population of splenic CD45^+^ CD11b^+^ F4/80^−^ MHC II^+^ (Figure 5E); CD45^+^ CD11b^+^ F4/80^+^ CD68^+^ (Figure 5F), and CD45^+^ CD11b^+^ F4/80^+^ CD68^+^ CD64^+^ (Figure 5G) showed no significant differences between the three groups.

To further analyze the systemic inflammatory phenotype, the concentration of serum C-reactive protein (CRP) was also detected. The results show that the inflammatory stimulation of DSS significantly increased the CRP concentration (*p* = 0.0006), but its concentration significantly decreased (*p* = 0.0428) following the prevention treatment with *L. rhamnosus* MS27 (Figure 5H).

### 2.8. Correlation Analysis Between the Microbial Species and Immune Cells via L. rhamnosus MS27 Treatment

In order to explore whether MS27 could modulate immune function via gut microbiota, correlation analysis between the microbial species and the immune cells was carried out. The results shown that *Lachnospiraceae_NK4A136_group* and *Colidextribacter* are significantly and positively related to the quantity percentage value of CD45^+^ CD11b^+^ F4/80^+^ CD68^+^ CD64^+^ cells (*p* < 0.05) and CD45^+^ CD11b^+^ F4/80^+^ CD68^+^ cells (*p* < 0.05) separately, while *Clostridium_sensu_stricto_1* is significantly negatively related to the quantity percentage value of CD45^+^ CD11b^+^ F4/80^+^ CD68^+^ CD64^+^ cells (*p* < 0.01), CD45^+^ CD11b^+^ F4/80^+^ CD68^+^ cells, and CD45^+^ CD11b^+^ F4/80^+^ CD68^+^ CD206^+^ cells (*p* < 0.05). *Turicibacter* is significantly negatively related to the quantity percentage value of CD45^+^ CD11b^+^ F4/80^+^ CD68^+^ CD64^+^ cells (*p* < 0.05) (Figure 6A).

Moreover, a co-linearity network analysis was executed to further exploring the distributions and the correlations between the microbial species and the spleen immune cells that to discover the key species who play a regulatory role. The results shown that *Turicibacter* and *Clostridium_sensu_stricto_1* are negatively related to the quantity percentage value of CD45^+^ CD11b^+^ F4/80^+^ CD68^+^ CD64^+^ cells, *Clostridium_sensu_stricto_1* is negatively related to the quantity percentage value of CD45^+^ CD11b^+^ F4/80^+^ CD68^+^ and CD45^+^ CD11b^+^ F4/80^+^ CD68^+^ CD206^+^ cells, however, *norank_f_Oscillospiraceae* is positively related to CD45^+^ CD11b^+^ F4/80^+^ CD68^+^ cells. In addition, *norank_o_Clostridia_UCG-014* is negatively related to the quantity percentage value of CD45^+^ CD11b^+^ MHC II^+^ cells and *norank_f_Oscillospiraceae* is positively related to the quantity percentage value of CD45^+^ CD11b^+^ cells (Figure 6B).

### 2.9. The Volatile Fatty Acid (VFA) and Fatty Acid (FA) Production of L. rhamnosus MS27

In order to examine if the metabolites produced by *L. rhamnosus* MS27 also play an important role, short-chain fatty acids (SCFAs) and FAs were detected, too. The results show that acetic acid is the main metabolite (Figure 6C) with the highest concentration (Table 3).

Meanwhile, the monounsaturated fatty acids oleic acid is the main fatty acid than others (Table 4).

## 3. Discussion

With the discovery of interactions between microorganisms and the host, it has become increasingly evident that the intestinal microbiota plays a crucial role in the pathogenesis and progression of inflammatory bowel disease (IBD), particularly for ulcerative colitis. However, the exact mechanisms underlying their mode of action remain to be fully elucidated [18,19]. Evidence from several strains indicates their critical role in regulating epithelial barrier function, achieved through the stimulation of tight junctions between epithelial cells, preservation of epithelial cell integrity, and induction of mucus production [20]. The capacity to modulate the host immune system is also considered to play a significant role, including the stimulation of antimicrobial defensins, interaction with local host immune cells to regulate cytokine production, and even providing benefits at distant sites [21]. Among these, *Lacticaseibacillus rhamnosus*, as a probiotic species, plays a significant role in supporting the treatment of inflammatory diseases; it achieves this by restoring the intestinal microbiota, improving intestinal barrier function, and reducing the levels of pro-inflammatory cytokines, among others. These effects are particularly relevant to the management of ulcerative colitis (UC) [14,22].

In terms of alleviating colitis induced by DSS, *L. rhamnosus* MS27 significantly reduced the decline of weight gain ratio; reduced the decline of colon length; reduced the amount of inflammatory cell infiltration; and showed a good effect in inhibiting colon inflammation. *L. rhamnosus* MS27 belongs to a group of porcine-derived probiotics isolated from the porcine saliva. As an animal model, a pig’s digestive physiology and metabolic processes are similar to that of a human’s. By applying novel molecular techniques to assess the intestinal microbiome, we are able to compare the similarity of representative bacterial species between pigs and humans [23]. Thus, pig-derived *L. rhamnosus* probiotics should be safe for humans, too. Due to differences in growth environment, the probiotics derived from pigs may have better antibacterial and anti-inflammatory properties. Our results show that *L. rhamnosus* MS27 is sensitive to carbenicillin, erythromycin, and tetracycline, and is moderately sensitive to chloramphenicol, clindamycin, and penicillin, and also has obvious inhibitory effects on common pathogenic bacteria, such as *Staphylococcus aureus*, *Escherichia coli,* and *Salmonella*. Its effects on inhibiting pathogens are consistent with the previous research results [24], which together further confirms its safety. Of course, its safety for human use still is worthy of further research and verification before it can be applied in clinical practice.

*L. rhamnosus* MS27 has a strong acid tolerance, and it can survive when the pH value drops to 3.23. Moreover, the number of viable bacteria remains at a high level within 24 h. These characteristics ensured that the oral formulation can smoothly pass through the stomach. However, *L. rhamnosus* MS27 has no pili, and whether it is a transient bacterial strain or a colonizing bacterial strain remains to be further investigated. In this study, analysis of the bacterial community of the intestinal chyme indicated that *L. rhamnosus* increased significantly in the LR+DSS group compared to the CON group and DSS group. It demonstrated *L. rhamnosus* MS27 could survive in the gastrointestinal tract, providing health benefits and proving the effectiveness of its oral administration.

*L. rhamnosus* MS27 significantly improved the intestinal microbial flora, which is regarded as an important way for it to exert beneficial effects. In this study, *L. rhamnosus* MS27 effectively influenced these microbial abundance changes, such as *Muribaculaceae*, *Akkermansia*, *Limosilactobacillus reuteri*, *Turicibacter*, and *Alistipes*. Among these, the *Muribaculaceae* bacteria is an anaerobic bacterium, and its quantity significantly increased with *L. rhamnosus* MS27 addition, which indirectly indicates that *L. rhamnosus* MS27 has beneficial effects on the recovery of barrier integrity in the colon.

To date, members of the family *Muribaculaceae* have attracted considerable attention due to their diverse roles in maintaining host health and are regarded as a promising “next-generation probiotic.” *Muribaculaceae* belongs to one of the dominant bacterial genera in the intestinal symbiotic flora and is mainly colonized in the intestinal tract of mammals. The *Muribaculaceae* family possesses the core metabolic capability to degrade a variety of complex polysaccharides. It can utilize dietary fiber interventions, such as inulin, resistant starch, and soluble fiber, to enhance the production of short-chain fatty acids. Additionally, members of this family are capable of encoding O-polysaccharide hydrolases and sialidases. The O-polysaccharidease is the most important enzyme for degrading mucin proteins, while the sialidase can cleave the sialic acid and sulfate residues at the terminal O-polysaccharide end of mucin proteins [25]. *Akkermansia muciniphila* is another famous “next-generation probiotic”. It can promote intestinal mucus secretion, helping to maintain the dynamic balance of intestinal mucus and regulating the intestinal mucosal barrier function, playing a crucial role in metabolic regulation and immune responses in the body. *Akkermansia muciniphila* can utilize mucin-derived sugars and degrade proteins of the intestinal mucus layer, such as mucin MUC2, as a carbon sources, thereby promoting host secretion of new mucus and enhancing gut barrier function [26,27]. *Limosilactobacillus reuteri* is a probiotic bacterium that resides in various human body sites, including the gastrointestinal tract, urinary tract, skin, and breast milk. *Limosilactobacillus reuteri* is capable of producing antibacterial molecules, including ethanol, reuterin, and organic acids, which inhibit the colonization of pathogenic microorganisms and contribute to the modulation of the host’s symbiotic microbial community [28]. *Limosilactobacillus reuteri* has been demonstrated to reduce the production of pro-inflammatory cytokines and enhance the differentiation and function of regulatory T cells. Moreover, it contributes to reinforcement of the intestinal barrier by improving the integrity of the colonic epithelial barrier through the upregulation of tight junction proteins, such as ZO-1, Occludin, and Claudin4 [29]. Over the past few decades, the abundance of *Limosilactobacillus reuteri* in humans has declined; this is associated with an increased incidence of inflammatory diseases during the same period. This correlation suggests that *Limosilactobacillus reuteri* may represent a promising strategy for the prevention and/or treatment of inflammatory diseases [30]. Clinical trials have further demonstrated that specific strains, such as *Limosilactobacillus reuteri* ATCC 6475, can alleviate intestinal inflammation in patients with ulcerative colitis [31]. *Turicibacter* participates in the modification of bile acids and host lipids metabolism differentially with strain-specific bsh genes [32]. *Turicibacter* can down-regulate the bile signaling expression in the liver [32]. *Alistipes* can produce hippuric acid to boost intestinal urate excretion via enhancing the binding of peroxisome-proliferator-activated receptor γ (PPARγ) to the promoter of ATP-binding cassette subfamily G member 2 (ABCG2) and maintain uric acid homeostasis [33]. These results collectively demonstrate that *L. rhamnosus* MS27 plays a role in modifying the structure and function of the intestinal microbiota to enhance the body’s response to inflammation, thereby restoring intestinal health.

Understanding the molecular mechanisms underlying the probiotic effects of *L. rhamnosus* MS27 is also essential for optimizing its use. The potential molecular mechanisms of *Lacticaseibacillus rhamnosus* on treatment of UC emphasize its ability to modulate the immune response, which is believed to be related to the bacterial pili, especially the spa CBA gene cluster encoding the pili [34]. The main strain of *Lacticaseibacillus rhamnosus* LGG is well-known and its surface is covered with pili, but *L. rhamnosus* MS27 has no pili structures, which means that *L. rhamnosus* MS27 has its own specific regulatory mechanism. In this study, we conducted a transcriptome–phenotype analysis to investigate the changes in the transcriptome after supplementing *L. rhamnosus* MS27 and the possible pathways in the interaction of immune-related proteins. Firstly, our results show that adding *L. rhamnosus* MS27 before the DSS processing, compared with the DSS alone, has a significant impact on the expression of two genes, *Igkv16-104* and *C1qtnf3*. Under the condition of inflammatory bowel disease, the expression of *Igkv16-104* is related to immune regulation and it should connect with *IL6* expression [35]. In inflammatory bowel diseases (IBDs) and other inflammatory disorders, the level of *IL-6* usually increases. Intestinal inflammation should stimulate macrophages/Th17 to secrete *IL-6*, which then activates the JAK-STAT3 pathway of B cell, thereby upregulating the Igκ enhancer expression (like *Igkv16-104*) and promoting antibody secretion, thereby exacerbating inflammation or tissue damage. Our results showed that the expression of *IL6* experienced a certain degree of decline after *L. rhamnosus* MS27 addition, but it did not drop significantly like *Igkv16-104* did. These results demonstrated that the decline of *Igkv16-104* should have another trigger mechanism.

C1q tumor necrosis factor related protein 3 (*C1qtnf3*) is a member of the C1q family that is upregulated under inflammatory conditions. Studies have indicated that *C1qtnf3* is involved in inflammatory responses and may play a role in the regulation of immune cell migration and activation, much like how *C1qtnf3* was upregulated during the remodeling of subcutaneous adipose tissue and promoted macrophage chemotaxis and M1-like polarization. This suggests that *C1qtnf3* may play a significant role in inflammation and immune regulation [36]. Following *L. rhamnosus* MS27 treatment, the expression of *36* was downregulated, which may be attributed to the effect of *L. rhamnosus* MS27 in regulating the balance of the intestinal microbiota and further contributing to the inhibition of inflammatory responses. *L. rhamnosus* MS27 may serve as a potential therapeutic strategy for the prevention and treatment of inflammatory diseases. However, the immunomodulatory role of *C1qtnf3* in the process of tissue remodeling merits further investigation in the future.

To investigate the potential mechanism of action of *L. rhamnosus* MS27, we further analyzed the genes that exhibited significant differential expressions between the *L. rhamnosus* MS27 + DSS treatment group and the DSS treatment group, as well as those that demonstrated 9*-protein interaction relationships. These are *H2Ab1*, *H2Eb1*, *H2k1*; *C1qa*, *C1qb*, *C1qc*, *C3ar1*, *C2*; *Xcl1*, *Cx3cr1*, *Cxcl12*, *Ccl22*, *Cxcl16*; and *Cd68*, *Mrc1*, *Csf1r*, *Tyrobp*, *Fcer1g*, which are different gene groups and mainly have immunity functions, antigen presentation, and immune cell mark, et al. The results show that preventive supplementation of *L. rhamnosus* MS27 also has a regulatory effect on the upregulation results of these genes induced solely by DSS, especially for *H2Ab1*, *H2Eb1*, and *H2k1*. They all belong to the MHC, corresponding to the *HLA* genes in humans. Herein, *H2-Ab1* and *H2-Eb1* are both belong to the MHC II class that present antigens to CD4+ T cells, however, *H2-Ab1* primarily regulates Th1/Th17 inflammatory responses, whereas *H2-Eb1* mainly influences immune tolerance. *H2-K1* belongs to MHC I class antigen and can be presented in CD8+ T cells to mediate cytotoxic immunity. *H2-Ab1* and *H2-Eb1* jointly form the MHC II class heterodimer (α + β), which affects the activation of CD4+ T cells [37], while H2-K1 independently presents endogenous antigens (such as viral proteins) to CD8+ T cells, regulating cellular immunity [38]. The upregulated expression of *H2-Ab1* exerts pro-inflammatory effects, such as promoting Th1/Th17 responses, thereby exacerbating autoimmune colitis in the DSS model [37]. When *H2-Eb1* upregulates expression, certain haplotypes (such as *H2-Eb1d*) promote the secretion of IL-17 and exacerbate intestinal inflammation [31], while the increased expression of *H2-K1* enhanced the killing ability of CD8+ T cells, exerting a protective effect in viral infections (such as influenza) and tumor immunity [39]. Based on these analyses, further examination of the impact of probiotics on immune phenotypes will be the next pressing question.

Disorders of the autoimmune system cause the intestinal lesions from UC to continue affecting other organs, leading to extraintestinal manifestations. After 14 days of probiotic treatment, its effects should not only work in the intestine but also reach the spleen via the lymphatic system or blood system, after which its intensity will weaken but the scope of its influence will expand. In order to further explore the mechanism of *L*. *rhamnosus* MS27, we further analyzed the changes in immune cells of the spleen.

The analysis results of splenic immune cells indicated that an increase happened in CD45^+^CD11b^+^MHCII^+^, but no changes happened in CD45^+^ CD11b^+^ F4/80^−^ MHC II^+^ cells and CD45^+^ CD11b^+^ F4/80^+^ CD68^+^ cells, and there is a non-significant decline in CD45^+^ CD11b^+^ F4/80^+^ CD68^+^ CD64^+^ cells. Overall, these results are consistent with the characteristics of “distal fine-tuning” of the spleen. Firstly, CD45^+^CD11b^+^MHCII^+^ is a mixed gate of innate immune cells derived from bone marrow, which expresses antigen-presenting molecules and mainly includes the classical CD11b^+^ dendritic cells (cDC2) and inflammatory monocytes (precursors of Mo-DCs that are upregulating MHC II). Via further analysis, the number of CD45^+^ CD11b^+^ F4/80^−^ MHC II^+^ cells, which are predominantly classical dendritic cells (cDC2), proved that cDC2 cells are stable, and while MS27 mainly affected the quantity of inflammatory monocytes, it perhaps plays a role in cytokine production, flexible antigen-presenting and innate-adaptive immune responses modulation, and tissue repair to inflammation resolution [14,40]. Secondly, CD45^+^ CD11b^+^ F4/80^+^ CD68^+^ CD64^+^ cells were gated as mature macrophage populations with high phagocytic and high Fc-γ receptor-mediated effector functions, mainly biased towards the M1 functional state [41]. Their decrease indirectly corroborates the improvement of the inflammatory state of the body, and the reduction in serum CRP concentration also supports this point. These results jointly supported the alleviation of the inflammatory response of MS27.

In combination with the regulatory effects of MS27 on intestinal microecology, its function of improving the immune system of the body may be associated with certain specific microorganisms. Among these, *Turicibacter* warrants particular attention. In our results, *Turicibacter* significantly increased in MS27 preventive treatment groups, in contrast with DSS treatment alone, and *Turicibacter* is significantly negatively related to CD45^+^ CD11b^+^ F4/80^+^ CD68^+^ CD64^+^. Thus, the anti-inflammatory effects of MS27 might be related to improvements in *Turicibacter* abundance. It can also produce conjugated linoleic acid to inhibit inflammasomes, and hence is helpful for inflammation alleviation [32]. We also explored whether MS27 could produce beneficial metabolic fatty acids to enhance its anti-inflammatory effects. Based on examination of the content of SCFAs and FAs of MS27, we found that acetic acid was its main volatile fatty acid product and oleic acid was the main fatty acid product. Here, acetic acid could work as a raw material for the synthesis of butyric acid to perform its beneficial function [42] and oleic acid is helpful for restoring the intestinal barrier to alleviate inflammation [43].

## 4. Materials and Methods

### 4.1. Strain Source and Identification

*L. rhamnosus* MS27 was isolated from oral saliva of Bama miniature pigs. Sterile swabs were used to collect oral mucus, placed in sterile centrifuge tubes, collected and filtered through sterile gauze, then inoculated at a concentration of 1% into 10 mL of *Lacticaseibacillus* selective medium. Cultivate anaerobically at 37 °C to 45 °C for 24 h to 36 h. Observe the colony morphology, pick out the oval-shaped strains with obvious calcium solubilization reaction, and conduct three streak purifications. After purification, sequence and identify the strains. Single colony was cultured on de Man, Rogosa, and Sharpe (MRS) solid medium plates at 37 °C for 24 h to purify them and this process was repeated for further purification. The Gram staining process proved that this strain is a Gram-positive, non-spore type intestinal symbiotic bacterium.

Using bacteria genomic DNA kit (CWBIO, Beijing, China) to extract total DNA of purified strains and amplifying them by PCR amplification using 27F (AGAGTTTGATCMTGGCTCAG) and 1492R (GGTTACCTTGTTACGACTT) universal primers [44] conducted in 50 μL reactions system, a product of 1500 base pairs in size was obtained. The sequencing results of the 16S rRNA of strain were compared with the NBCI database and the 16S rRNA gene sequences with similar homologies were selected from GenBank. Furthermore, the phylogenetic tree was constructed using MEGA 11.0. The sequence alignment was constructed using the Neighbor-joining method, and the self-expansion method (Bootstrap) was used for verification. The number of repetitions was 2000 times, and confirmed that this strain is a new strain of *Lacticaseibacillus rhamnosus*; it was named *Lacticaseibacillus rhamnosus* MS27 (*L. rhamnosus* MS27). It was stored in the China General Microbiological Culture Collection Center (CGMCC No. 27412).

### 4.2. Growth and Survival Curves, pH Tolerance Curves

Activated purified strains were incubated in MRS broth at 37 °C and then were tested on OD600 nm values to express viable bacteria numbers every 2 h within 24 h. Furthermore, they were incubated with different pH (0, 2.5, 3.0, and 4.0) of sterile phosphate-buffered saline (PBS) (Gibco, Brooklyn, NY, USA) at 37 °C for 0, 1, 2, 3, 4 h to quantify viable bacterial counts using the plate colony counting method. The viable bacterial count at 0 h was used as the control to calculate the survival rate of *L. rhamnosus* MS27 in different pH conditions. This method follows our previous procedure [45]. Prism 10.0 was carried out to analyze data and create growth curves, survival curves, and pH curves.

### 4.3. Biochemical Identification and Antibiotic Sensitivity Analysis

Prepare *L. rhamnosus* MS27 that has been purified and can grow into single colonies and take single colonies from the plate with an inoculation needle and inoculate them into the biochemical identification tubes of *Lacticaseibacillus* (No. SHBG13, Hopebio, Qingdao, China). Cultivate them according to the methods in the manual and determine the reaction conditions of the MS27 strain to sucrose, lactose, maltose, inulin, raffinose, esculin, salicin, and mannitol, respectively.

The drug resistances of the strains were evaluated by conducting drug sensitivity analysis using representative types of antibiotics. The main steps are as follows: penicillins (penicillin, carbenicillin), macrolides (erythromycin), tetracyclines (tetracycline), aminoglycosides (chloramphenicol), and lincosamides (clindamycin). Logarithmic growth phase *L. rhamnosus* MS27 was inoculated at a 1% ratio onto MRS solid medium at 45 °C to 55 °C and left to stand for 20 min. After the medium solidified, the drug sensitivity test paper was picked up with forceps and placed in the center of the solid plate, gently pressed down, and then incubated at 37 °C for 48 h. The growth of the strains was observed. Inhibition zone diameters of less than 15 mm were labeled as resistant (R), those between 16 mm and 20 mm were moderately sensitive (I), and those greater than 20 mm were sensitive (S).

### 4.4. Antibacterial Property of Pathogenic Bacteria

To explore the ability to inhibit pathogenic bacteria in vitro, pick out *L. rhamnosus* MS27 from the solid plate, inoculate it into MRS liquid medium, and ferment and culture at 37 °C for 36 h at 100 r/min. The effects of bacterial liquid, fermentation supernatant, and bacterial cells on common pathogenic bacteria in livestock and poultry were detected, respectively. The bacterial liquid was centrifuged at 4 °C and 5000 rpm for 10 min, and the supernatant was obtained after filtering the membrane. The bacterial cells were precipitated and re-resuspended with an equal volume of sterile physiological saline. *Escherichia coli K88*, *Staphylococcus aureus* ATCC 25923, and *Salmonella typhimurium* SL1344 growing on solid plates were picked and inoculated into 30 mL of sterilized liquid medium. After culturing at 37 °C for 12 h, the concentration was adjusted with sterile normal saline, and the pathogenic bacteria were diluted to 1.0 × 10^8^ CFU/mL for later use. Place Oxford cups on plain AGAR plates. Cool the sterilized solid LB mehhdium to 45–55 °C. Mix 20 mL of solid LB medium +200 μL of diluted pathogen solution evenly and pour it onto the plates. After cooling and solidification, take out the Oxford cups and add 150 μL of sample to each well. Let it stand at 4 °C for 2–4 h and then incubate at 37 °C for 24 h.

### 4.5. Scanning Electron Microscope (SEM) Observation

We performed samples according to the China Agricultural University Bio-ultrastructure analysis Lab’s rules for bacteria sample preparation. After liquid enrichment culture of the *L. rhamnosus* MS27 strain, we removed the supernatant carefully by 2000 rpm, 5 min centrifugation. Resuspended the bacteria in 2 mL 2.5% glutaraldehyde to fix them for at least 2 h. Then we transferred them to a 4 °C environment for storage. Before observation, the fixed *L. rhamnosus* MS27 strain samples need to be vacuum-dried and dehydrated, then the dried powder should be transferred to the sample platform with conductive glue. Gold should be sprayed onto the samples by sputtering to enhance their conductivity. After that, the prepared samples were observed in S3400N SEM (Hitachi, Ltd., Tokyo, Japan).

### 4.6. Animal Experimental Protocol and Sample Collection

The C57BL/6 mice used in this trial are laboratory animals from Beijing HFK Bio-Technology. Co., Ltd, Beijng, China. Before the formal trial begins, a three-day acclimatization period is implemented to enable the animals to adapt to the environment. After that, the trial period is 14 days and the gavage was performed every 2 days. Eight-week-old 24 C57BL/6 male mice were randomly divided into three groups, as follows: the control group (CON, gavage with 200 µL normal saline every 2 days on 1–14 d), Dextran Sulfate Sodium Salt (DSS) group (gavage with 200 µL normal saline every 2 days on 1–14 d and drink 3% DSS on 7–14 d), *and L. rhamnosus* MS27+DSS group (LR+DSS, gavage with 200 µL 1 × 10^8^ CFU/mL *L. rhamnosus* MS27 every 2 days from 1 to 14 d and drink 3% DSS on 7–14 d). Each group has eight replicates.

During the experimental time, daily weight was recorded every day. Based on these data, weight gain and the weight gain ratio (weight gain to the original body weight) were calculated. All mice were selected from each group for sampling on the 14th day. Obtain the weight of the liver and calculate the liver index (liver weight to body weight). Liver Bloods were collected and centrifuged at low temperature to obtain serum, then were stored at −20 °C for subsequent experiments. Separated total colon to measure its length and then collected colon sample formalin fixation for histological sections and HE staining. In addition, keeping collected colon tissue with liquid nitrogen treated immediately for RNA extraction and sequencing. Collected colon chyme in 1.5 mL sterile centrifuge tube and stored at −80 °C for subsequent 16S rRNA and metabolomic sequencing. All experimental procedures complied with relevant management regulations of Animal Protection and Utilization Committee of China Agricultural University (CAU20171015-3).

### 4.7. RNA Extraction, Reverse Transcription, and Gene Expression Analysis by Quantitative Real-Time PCR

Total colon tissue RNA was extracted using TRIzol^®^ Reagent following the manufacturer’s instructions. Subsequently, RNA was quantified using the ND-2000 spectrophotometer (NanoDrop Technologies, Wilmington, DE, USA). Only high-quality RNA samples were used with OD260/280 ratios ranging from 1.8 to 2.2, RIN values greater than 6.5, and concentrations exceeding 10 ng/μL.

Subsequently, RNA was reverse transcribed into cDNA at 42 °C for 60 min using random hexamer primers and the RevertAid RT Kit (Cat. No. EP0441, Thermo Fisher Scientific, Wilmington, DE, USA), following the manufacturer’s instructions. Real-time PCR was performed to assess the relative expression levels of the target genes. RT-qPCR reactions were performed with tenfold-diluted cDNA using the PowerUp SYBR Green Master Mix (No. Q712, Vazyme, Nanjing, China). Each reaction was prepared in a 20 μL reaction mixture containing 10 μL of Power SYBR Green master mix, 0.3 μM of each primer adjusted to the designated final concentration, and 5 μL of diluted cDNA. The thermal cycling protocol was as follows: initial incubation at 50 °C for 2 min, followed by denaturation at 95 °C for 2 min, then 40 cycles of denaturation at 95 °C for 15 s, and annealing/extension at 60 °C for 1 min with fluorescence detection. A final melt curve analysis was performed, consisting of incubation at 95 °C for 15 s, cooling to 60 °C for 1 min, and a temperature ramp to 95 °C at a rate of 0.15 °C/s.

### 4.8. 16S rRNA Microbiome Analysis (Second-Generation OTU)

Total microbial genomic DNA was extracted from fecal samples using the E.Z.N.A.^®^ DNA Kit (Omega Bio-tek, Norcross, GA, USA) following the manufacturer’s instructions. The quality and concentration of the extracted DNA were assessed by 1.0% agarose gel electrophoresis and using a NanoDrop 2000 spectrophotometer (Thermo Scientific, DE, USA), and the DNA was stored at −80 °C until further use. The hypervariable V3–V4 region of the bacterial 16S rRNA gene was amplified using the primer pairs 338F (5′-ACTCCTACGGGAGGCAGCAG-3′) and 806R (5′-GGACTACHVGGGTWTCTAAT-3′) on a T100 Thermal Cycler PCR system (BIO-RAD, Hercules, CA, USA). The PCR reaction mixture consisted of 4 μL of 5× Fast Pfu buffer, 2 μL of 2.5 mM dNTPs, 0.8 μL of each primer (5 μM), 0.4 μL of Fast Pfu polymerase, 10 ng of template DNA, and ddH_2_O to a final volume of 20 µL. The PCR amplification program included an initial denaturation at 95 °C for 3 min, followed by 27 cycles of denaturation at 95 °C for 30 s, annealing at 55 °C for 30 s, and extension at 72 °C for 45 s, with a final extension at 72 °C for 10 min, and a holding step at 4 °C. The resulting PCR products were excised from a 2% agarose gel, purified using the PCR Clean-Up Kit (YuHua, Shanghai, China) according to the manufacturer’s instructions, and quantified using a Qubit 4.0 fluorometer (Thermo Fisher Scientific, Wilmington, DE, USA).

During the Illumina sequencing step, purified amplicons were pooled in equimolar concentrations and subjected to paired-end sequencing on an Illumina Nextseq 2000 platform (Illumina, San Diego, CA, USA) following the standard protocols provided by Majorbio Bio-Pharm Technology Co., Ltd. (Shanghai, China). The raw sequencing reads were submitted to the NCBI Sequence Read Archive (SRA) database under Accession Number PRJNA1276053. Raw FASTQ files were de-multiplexed using a custom Perl script, quality-filtered with fastp version 0.19.6 [46], and merged using FLASH version 1.2.7 [47]. Subsequently, the processed sequences were clustered into operational taxonomic units (OTUs) at a 97% sequence similarity threshold using UPARSE 11.0 and UNOISE3 11.0 [48,49]. The most abundant sequence within each OTU was selected as the representative sequence.

Bioinformatic analysis of the gut microbiota was carried out using the Majorbio Cloud platform (https://cloud.majorbio.com, 30 March 2025) [50]. Based on the OTU information, rarefaction curves and alpha diversity indices, including the number of observed OTUs, Chao1 richness estimator, Shannon diversity index, and Good’s coverage, were calculated using Mothur v1.30.1 [51]. The similarity among microbial communities in different samples was assessed using principal coordinate analysis (PCoA) based on Bray–Curtis dissimilarity with the Vegan v2.5-3 package. The PERMANOVA test was conducted to evaluate the proportion of variation attributable to treatment effects and its statistical significance, using the same package. Linear discriminant analysis (LDA) effect size (LEfSe) [52] was applied to identify bacterial taxa (from phylum to genus level) that were significantly enriched across different groups (LDA score > 2, *p* < 0.05).

### 4.9. Eukaryotes Reference Transcriptome Analysis

After RNA extraction and library preparation, the RNA-seq transcriptome library was prepared following SMART-Seq_V4 Ultra Low Input RNA Kit for Sequencing from Clontech (San Diego, CA, USA) using 10 ng of total RNA. Reverse transcription (one-strand synthesis) was performed first, and RNA with a poly Atail (major mRNA) was reverse transcribed using Oligo (dT) primer. Three cytosine (C) residues are added to the 3′ end of the cDNA strand because of the use of a special active reverse transcriptase (MMLVRT) for reverse transcription. Two strands of cDNA were synthesized with the use of TSO (template-switching oligo) primers, thereby replacing RNA complementary to one strand of cDNA. Then, the cDNA was expanded to ng level by PCR amplification. DNA disruption using a modified, highly active Tn5 transposase was performed while the linker was added to both ends of the cDNA. After the last PCR amplification, the machine was ready for sequencing. Following quantification using Qubit 4.0, the sequencing libraries were prepared and subjected to paired-end sequencing (PE150) on either the NovaSeq X Plus platform using the NovaSeq Reagent Kit or the DNBSEQ-T7 platform using the DNBSEQ-T7 RS Reagent Kit (FCL PE150), version 3.0.

The raw paired-end reads were trimmed and subjected to quality control using FastQC [49] with default parameters. Subsequently, the cleaned reads were aligned to the reference genome in strand-specific mode using HISAT2 [53]. The mapped reads from each sample were assembled using StringTie [54] through a reference-based approach. To identify differentially expressed genes (DEGs) between the two sample groups, transcript expression levels were calculated based on the transcripts per million (TPM) method. Gene abundance quantification was performed using RSEM [55]. Differential expression analysis was carried out using either DESeq2 [56] or DEGseq [57], as appropriate. DEGs with |log2FC| ≥ 1 and FDR < 0.05 (DESeq2) or FDR < 0.001 (DEGseq) were considered to be significantly different expressed genes. In addition, functional enrichment analysis including GO and KEGG were performed to identify which DEGs were significantly enriched in GO terms and metabolic pathways at Bonferroni-corrected *p*-value < 0.05 compared with the whole-transcriptome background. GO functional enrichment analysis and KEGG pathway enrichment analysis were performed using Goatools and the Python (3.10) SciPy package, respectively. All RNA-seq data have been submitted to the GSA database under the following accession numbers: PRJNA1283609.

### 4.10. Flow Cytometry Experiment

Following the separation of the spleen, it was aseptically minced into small fragments using sterile scissors and subsequently dissociated through a 70 μm cell strainer with the aid of sterile forceps. A single-cell suspension was generated using a mouse spleen lymphocyte isolation kit (Cat#P8860, Beijing Solarbio Science & Technology Co., Ltd., Beijing, China) according to the manufacturer’s protocol. The cell concentration of the suspension was adjusted to 10^8^–10^9^ cells/mL based on the spleen volume.

The CytoPeak R1 3-laser 14-color flow cytometer (Dakewe Biotech Co., Ltd., Shenzhen, China) was used to detect the CD 45 (PerCP/Cyanine5.5 Anti-Mouse CD45, Cat#E-AB-F1136J, Elabscience, Wuhan, China), CD64 (PE/Cyanine7 Anti-Mouse CD64/FcγRI Antibody[X54-5/7.1], Cat#E-AB-F1186UH, Elabscience, China), F4/80 (Elab Fluor^®^ Violet 610 Anti-Mouse F4/80 Antibody[CI:A3-1], Cat#E-AB-F0995T, Elabscience, China), MHC II (APC Anti-Mouse MHC II Antibody[M5/114], Cat#E-AB-F0990UE, Elabscience, China), CD206 (FITC Anti-Mouse CD206/MMR Antibody[C068C2], Cat#E-AB-F1135C, Elabscience, China), CD19 (Elab Fluor^®^ Violet 450 Anti-Mouse CD19 Antibody, Cat#E-AB-F0986UQ, Elabscience, China), CD11b (PE CD11b anti-mouse, Cat#E-AB-F1081D, Elabscience, China) and CD68 (CoraLite^®^ Plus 750 Anti-Mouse CD68 Rabbit Recombinant Antibody, Cat#CL750-98029, Proteintech, Wuhan, China).

### 4.11. Detection of Volatile Fatty Acids and Fatty Acids

After 12 h of *L. rhamnosus* MS27 culturing, the supernatant was taken after centrifugation at 2000 rpm to detect the SCFAs via an iron chromatographic method as previously reported [45]. 1 mL of supernatant was diluted with dd H_2_O (1:50) and filtered with 0.22 μm filter. Finally, it was injected into a high-performance ion chromatograph (ICS 3000 Dionex, Sunnyvale, CA, USA) to detect the content of acetic acid, propionic acid, isobutyric acid, butyric acid, isovaleric acid, valeric acid, oleic acid, stearic acid, palmitic acid, and octanoic acid.

### 4.12. Detection of Serum CRP Concentration

Serum CRP concentration was measured by high-sensitivity ELISA (Cat# SEKM-0059, Beijing Solarbio Science & Technology Co., Ltd.) and expressed as µg L^−1^. The limit of detection was 0.1 mg L^−1^. *n* = 8.

### 4.13. Statistical Analysis

All the graphs and analyses were carried out by GraphPad prism 10.0. The results were shown on mean ± SEM. Statistical significance was considered at *p* < 0.05.

## 5. Conclusions

As a novel strain, *L. rhamnosus* MS27 exhibits unique characteristics in lactic acid utilization and acetic acid and oleic acid production. Moreover, we additionally did an exploratory analysis of the microbiota–immune axis elucidation, which highlighted that MS27 improves the abundance of *Turicibacter*, and demonstrated its negative correlation with pro-inflammatory macrophage subsets (CD45^+^CD11b^+^F4/80^+^CD68^+^CD64^+^), suggesting a novel microbiota–immune pathway for systemic immune homeostasis. These characteristics of *L. rhamnosus* MS27 may collaboratively contribute to the alleviation of intestinal inflammation.

## Figures and Tables

**Figure 1 ijms-26-11397-f001:**
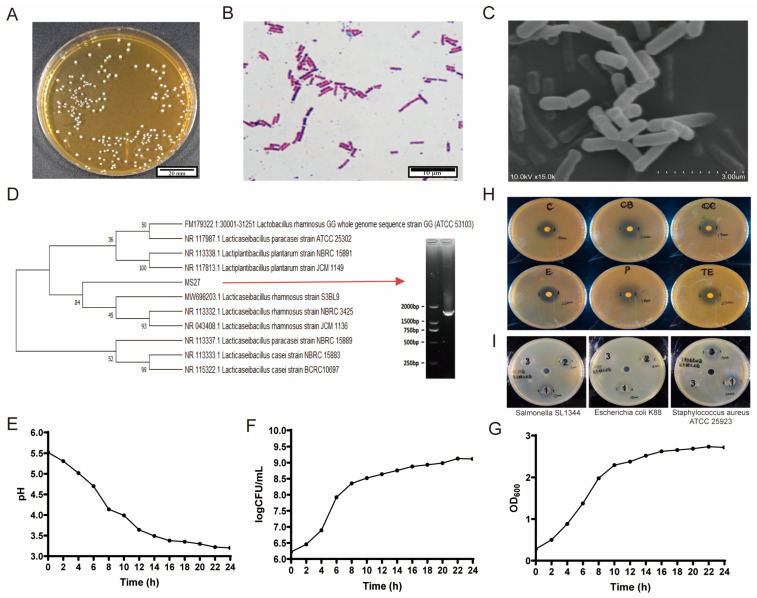
Identification of *L. rhamnosus* MS27. (**A**) Colony morphology of *L. rhamnosus* MS27, bar = 20 mm. (**B**) Gram staining of *L. rhamnosus* MS27, bar = 10 μm. (**C**) Scanning electron microscope of *L. rhamnosus* MS27 morphology, enlarged 15,000 times, 3 μm. (**D**) The phylogenetic tree of *L. rhamnosus* MS27. (**E**) Acid tolerance of *L. rhamnosus* MS27. (**F**) The growth curve of *L. rhamnosus* MS27. (**G**) Bacterial count curve of *L. rhamnosus* MS27. (**H**) For the antibiotic sensitivity of *L. rhamnosus* MS27, C is chloramphenicol; CB is carbenicillin; CC is clindamycin; E is erythromycin; TE is tetracycline; and P is penicillin. (**I**) The active ingredients of *L. rhamnosus* MS27: 1 is the bacterial solution; 2 is the fermentation supernatant; and 3 is the bacterial cells. The pathogenic bacteria contained on the culture media are *Salmonella* SL1344, *Escherichia coli K88,* and *Staphylococcus aureus* ATCC 25923.

**Figure 2 ijms-26-11397-f002:**
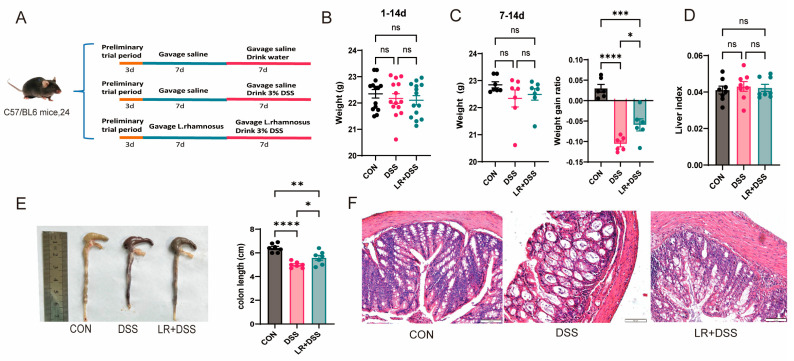
(**A**) Experimental design of animal trial. (**B**) The body weight change in mice for the entire period of 1–14 d and the body weight change after drinking 3% DSS. *n* = 14. (**C**) The body weight changes and the ratio body weight gain to the body weight of 7–14 d in mice after DSS treatment. *n* = 7. (**D**) Relative organ weight of liver to body weight. (**E**) The measurement results of colon length on the 14th day. *n* = 7; (**F**) Histopathological H&E staining of the colon tissues in mice at the end of the trial. The magnification scale is 100 μm. All numerical values are denoted in terms of mean ± SEM; ns, *p* > 0.05; *, *p* < 0.0.5; **, *p* < 0.01; ***, *p* < 0.001; ****, *p* < 0.0001.

**Figure 3 ijms-26-11397-f003:**
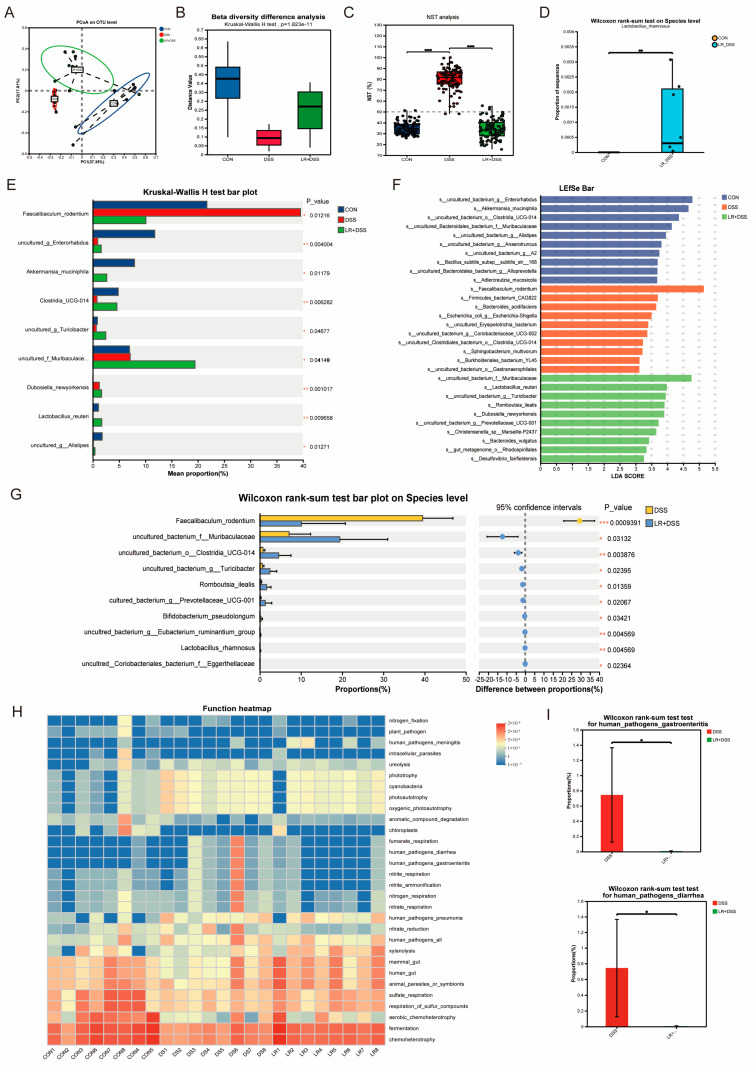
(**A**) PCoA analysis to explore the differences in biodiversity of community compositions among different groups. A two-dimensional visual scatter plot displays degrees of similarity and difference for the different communities among groups. The distance between samples reflects the degree of clustering and dispersion of the sample communities. This analysis result is statistically analyzed using ANOSIM/adonis/PERMAVONA, presenting the significant differences in the community of the control group and the treatment group. The scale of the *X*-axis and *Y*-axis is in terms of relative distance, and has no practical meaning; points of different colors or shapes represent samples from the different groups. The closer the two sample points are, the more similar the species composition of the two samples. (**B**) Beta diversity analysis for inter-group differences between samples. Different distance methods calculate the distance matrices, which are the distance indices of beta diversity. Statistical tests are conducted on the distance differences between different groups to reflect the degree of sample dispersion within the groups. And this result is consistent with the inter-group variation form of the Shannon index in alpha diversity. (**C**) NST analysis is undertaken to uncover the underlying mechanisms of community formation and assess and predict the responses of communities to environmental changes. This is based on the algorithm of Jaccard distance, as well as the FDR multiple testing correction method. The horizontal axis represents the sample groups; the vertical axis represents the NST values; and the dotted line in the figure is the threshold for the division between certainty and randomness, indicating the significance of the differences between groups. The box plot shows the maximum value, upper quartile value, median value, lower quartile value, and minimum value. (**D**) Single-species comparisons of *L. rhamnosus* between the DSS group and the LR+DSS group. FDR multiple testing correction was adopted; the confidence level of bootstrap was calculated with 0.95. (**E**) Inter-group significance test analysis: The bar chart shows the average relative abundance differences in the same species among different groups, and it indicates whether the differences are significant, with the different colored bars representing different groups, and the rightmost part is the *p*-value. (**F**) LEfSe multi-level species difference discrimination analysis. Bar chart shows the LDA values of different distinct species, visually presenting the influence magnitude of the characteristic species identified in different groups on the differences. The LDA discriminant bar chart statistically shows the significant microbial groups in multiple groups, obtained through LDA, with a higher LDA score indicating a greater influence of the species abundance on the difference effect. (**G**) The inter-group difference test is used to analyze whether there are differences in the microbial composition of ester bonds between LR+DSS group and DSS group, and to identify the microorganisms with significant differences. Testing method picks up Wilcoxon rank sum test. (**H**) FAPROTAX function prediction heat map is used to display the distribution of functional abundance in different samples, which intuitively reflects the distribution of the main dominant functions in different samples. The horizontal axis represents the group name, the vertical axis represents the function name, and the abundance changes of different functions in the samples are shown through the color gradient of the color blocks. The values represented by the color gradient on the right side of the figure are as shown. (**I**) Single-function comparison chart of different groups under the FAPROTAX function prediction heat map. The bar chart is used to display the percentage of a certain function in different grouped samples. The horizontal axis represents the group name, the vertical axis represents the percentage of the abundance of the function in different samples, and different colors represent different groups. The rightmost part shows the *p*-value, * 0.01 < *p* ≤ 0.05, ** 0.001 < *p* ≤ 0.01, *** *p* ≤ 0.001.

**Figure 4 ijms-26-11397-f004:**
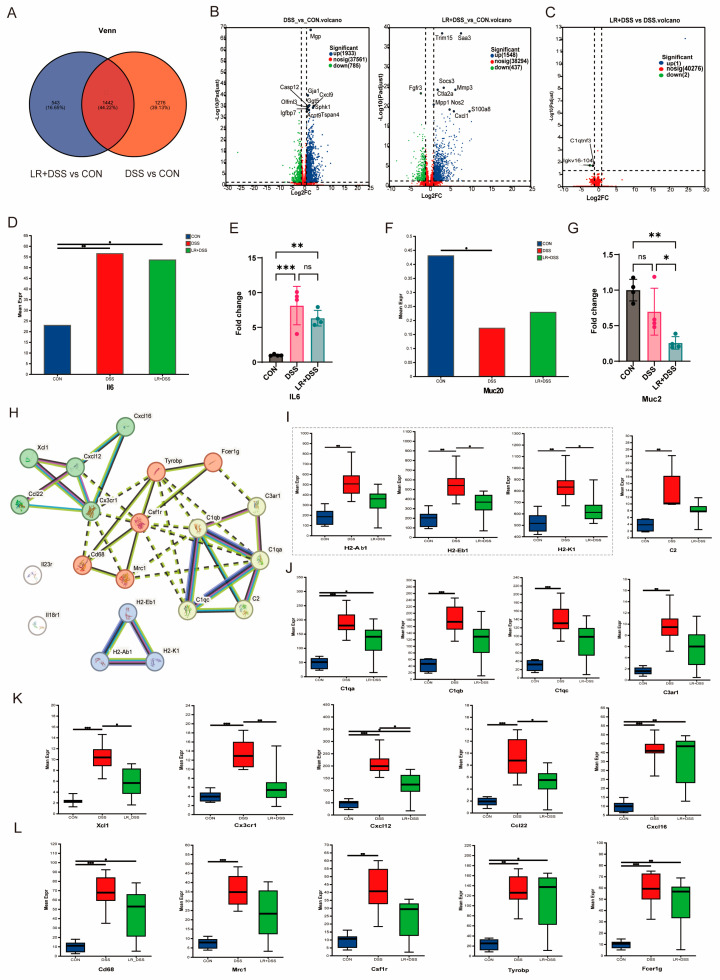
(**A**) VENN analysis results. Red-colored circles represent different gene sets of DSS group vs. CON group, and blue-colored circles represent different gene sets of LR+DSS group vs. CON group. The numbers inside the circles represent the total number of shared and unique genes/transcripts among different gene sets. (**B**) Statistical volcano chart of gene difference in expression quantity representing the DSS group vs. CON group and LR+DSS group vs. CON group. The top 10 genes with significant differences are shown as dots with their name. (**C**) Statistical volcano chart of gene difference in expression quantity representing the LR+DSS group vs. DSS group. The horizontal axis represents different comparison groups of differences, while the vertical axis represents the corresponding numbers of upregulated genes/transcripts. Blue dots indicate upregulated expression and green ones indicate down-regulated expression. Red dots indicate genes with no significant difference. The *x*-axis represents the fold difference in expression of the gene/transcript between the two samples, which is the value obtained by dividing the expression level of the treated sample by that of the control sample. The *y*-axis represents the statistical test value of the difference in gene expression level, namely the *p*-value. The larger the −log10 (*p*-value), the more significant the expression difference. Both the x- and *y*-axis values have been logarithmically transformed. Each point in the graph represents a specific gene. The closer the dots are to the edge, the more significant the expression difference. (**D**) The expression level of interleukin 6 (*IL 6*): Sample number *n* = 6. (**E**) The relative expression level of *IL6* in colon tissue. *n* = 4. (**F**) The expression level of mucin 20 (*Muc20*), *n* = 6. (**G**) The relative expression level of *Muc2* in colon tissue. *n* = 4. (**H**) PPI network map; the default value is ≥0.7. MCL clustering algorithm. Number of nodes is 20, number of edges is 39, average node degree is 3.9, avg. local clustering coefficient is 0.628, expected number of edges is 1, and PPI enrichment *p*-value < 1.0 × 10^−16^. Colored nodes are the specific queried proteins and first shell of interactors; the white nodes are the proteins of second shell of interactors. The node with filled means a 3D structure of this protein is known or predicted. The line color indicates the type of interaction evidence. Dotted line means the edge between clusters. (**I**) The expression level of *H2Ab1*, *H2Eb1*, *H2k1*. *n* = 6. (**J**) The expression level of *C1qa*, *C1qb*, *C1qc, C3ar1*, *C2*. *n* = 6. (**K**) The expression level of *Xcl1*, *Cx3cr1*, *Cxcl12*, *Ccl22*, *Cxcl16*. *n* = 6. (**L**) The expression level of *Cd68*, *Mrc1*, *Csf1r*, *Tyrobp*, *Fcer1g*. *n* = 6. All numerical values are denoted in terms of mean ± SEM; ns, *p* > 0.05; *, *p* < 0.05; **, *p* < 0.01; ***, *p* < 0.001.

**Figure 5 ijms-26-11397-f005:**
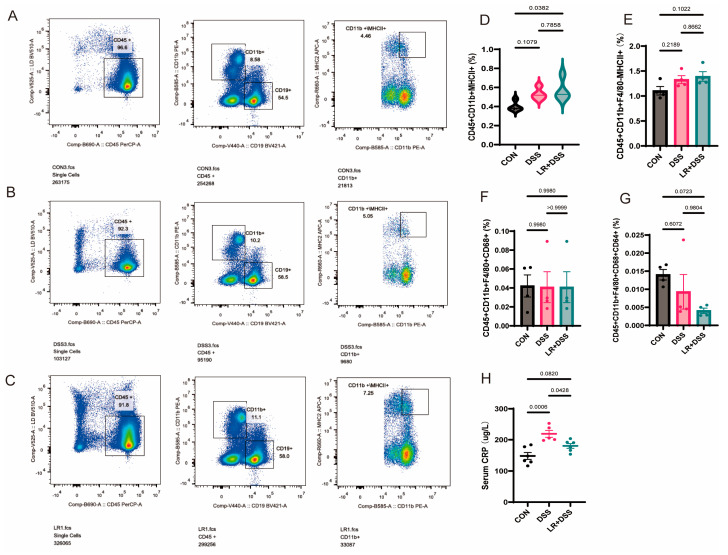
(**A**) The quantity percentage of CD45^+^, CD11b^+^, and MHC II^+^ cells in control group, *n* = 4. (**B**,**C**) The quantity percentage of CD45^+^, CD11b^+^, and MHC II^+^ cells in DSS and LR+DSS treatment groups; *n* = 4 for each treatment. All samples were processed using the same gating strategy. The quantity percentage of non-viable cells is on average, 8.357 ± 0.646 (%). (**D**) The quantity percentage values of CD45^+^ CD11b^+^ MHC II^+^ cells in three groups shown with violin plot. *p* < 0.05 indicates a significant difference. *n* = 4. (**E**) The quantity percentage values of CD45^+^ CD11b^+^ F4/80^−^ MHC II^+^ cells in three groups, *n* = 4. (**F**) The quantity percentage values of CD45^+^ CD11b^+^ F4/80^+^ CD68^+^ cells in three groups, *n* = 4. (**G**) The quantity percentage values of CD45^+^ CD11b^+^ F4/80^+^ CD68^+^ CD64^+^ cells in three groups, *n* = 4. (**H**) Serum CRP concentration in three groups; *p* < 0.05 means a significant difference, *n* = 8.

**Figure 6 ijms-26-11397-f006:**
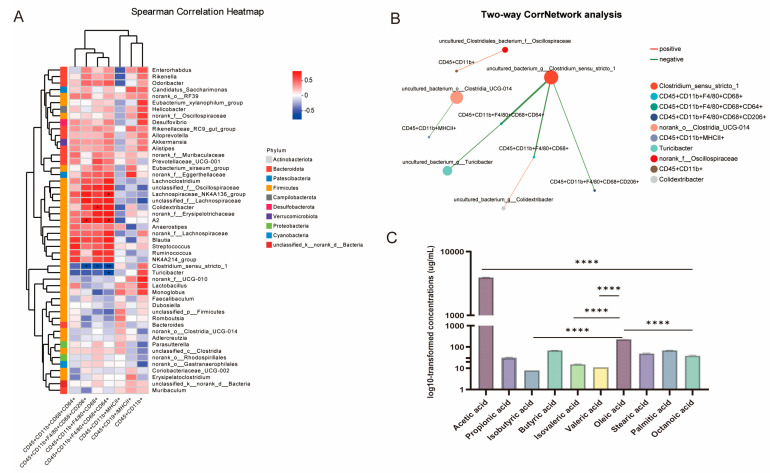
(**A**) The spearman correlation heat map shows the correlation between species and environmental factors (splenic immune cells). The *X*-axis and *Y*-axis represent environmental factors and species, respectively. The correlation R value and *p*-value are obtained through calculation. The R value is displayed in different colors in the graph, and the *p*-value is marked with * if it is less than 0.05. *, 0.01 < *p* ≤ 0.05; **, 0.001 < *p* ≤ 0.01. (**B**) The two-factor correlation network diagram focuses on analyzing the correlation between microbial species and environmental factors (splenic immune cells), facilitating the understanding of the interaction between them. The top 50 species are selected, and the Spearman rank correlation and other correlation coefficients between species are calculated to reflect the correlation among species, and species with *p* < 0.05 are displayed. The size of the nodes represents the abundance of the species, and different colors represent different species. The lines in red represent a positive correlation and those in green represent a negative correlation. The thickness of the lines represents the magnitude of the correlation coefficient, and the thicker the line, the greater the correlation. The presence of more lines indicates a closer connection between the nodes. (**C**) The volatile fatty acid and fatty acid production of *L. rhamnosus* MS27. ****, *p* ≤ 0.0001.

**Table 1 ijms-26-11397-t001:** Fermented carbohydrate comparison of *L. rhamnosus* MS27 and other *Lacticaseibacillus rhamnosus* strains.

Items	*L. rhamnosus* MS27	*L. rhamnosus* LGG [15]	*L. rhamnosus* NBRC3425 [16]	*L. rhamnosus* JCM1136 [17]
Sucrose	+	+	+	+
Lactose	+	−	−	−
Maltose	+	+	+	+
Inulin	+	+	/	/
Synanthrin	+	+	+	+
Raffinose	−	−	/	/
Esculin	+	+	/	+
Salicin	+	/	/	/
Mannitol	+	+	+	+
Sorbitol	+	/	/	+

Note: +, represents a positive effect; − represents a negative effect; / means that there is no data to support the result.

**Table 2 ijms-26-11397-t002:** Antibiotic susceptibility table of *L. rhamnosus* MS27.

Antibiotics	Abbreviation	IZD (mm)	Sensibility
Chloramphenicol	C	18.3 ± 0.06	I
Carbenicillin	CB	21.4 ± 0.05	S
Clindamycin	CC	19.1 ± 0.03	I
Erythromycin	E	22.0 ± 0.03	S
Tetracycline	TE	23.1 ± 0.01	S
Penicillin	P	17.2 ± 0.07	I

**Table 3 ijms-26-11397-t003:** The volatile fatty acid production of *L*. *rhamnosus* MS27.

Acetic Acid (μg/mL)	Propionic Acid (μg/mL)	Isobutyric Acid (μg/mL)	Butyric Acid (μg/mL)	Isovaleric Acid (μg/mL)	Valeric Acid (μg/mL)
3944.11 ± 2.13	31.55 ± 0.49	7.95 ± 0.01	69.58 ± 0.20	14.82 ± 0.19	11.11 ± 0.06

Note: All numerical values adenotes with mean ± SEM; *n* = 3.

**Table 4 ijms-26-11397-t004:** The fatty acids production of *L. rhamnosus* MS27.

Oleic Acid (μg/mL)	Stearic Acid (μg/mL)	Palmitic Acid (μg/mL)	Octanoic Acid (μg/mL)
230.06 ± 0.10	50.07 ± 0.05	70.09 ± 0.04	40.12 ± 0.05

Note: All numerical values are denoted in terms of mean ± SEM; *n* = 3.

## Data Availability

The raw sequencing reads of 16S rRNA Microbiome analysis were submitted to the NCBI Sequence Read Archive (SRA) database under Accession Number PRJNA1276053. The RNA-seq transcriptome All RNA-seq data have been submitted to the GSA database under the following accession numbers: PRJNA1283609.

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
