# Peer review of "Lacticaseibacillus rhamnosus* MS27 Potentially Prevents Ulcerative Colitis Through Modulation of Gut Microbiota"

_ijms, 2025, doi:10.3390/ijms262311397_

Round 1
Reviewer 1 Report
Comments and Suggestions for Authors
In this study, Zhang et al., investigated the effects of a purified Lacticaseibacillus rhamnosus MS27 on modulating DSS-induced intestinal inflammation. Integrating the methods including microbial sensitive test, 16S rRNA sequencing, and transcriptomic analysis, authors demonstrated that the L.rhamnosus MS27 could potentially regulates immune response and prevent ulcerative colitis via modulating gut microbiota. However, whether and how L.rhamnosus MS27 prevents inflammation via gut microbiota are still unclear. Below are comments.
Major points:
- In the context of introduction (line 50-52), the authors state that conventional therapies such as 5-aminosalicylic acid, glucocorticoids, and et al result in side effects in UC therapy. As such, it’s highly recommended to explain – a. what kind of side effects are basically induced by the above therapies; b. why those therapies causes side effects in UC patients, whether the effects of the above are relevant to gut microbiota. At least, there should be references cited here.
- Figure 1 characterized the feature of L.rhamnosus MS27 via colony, pH, growth curve, acid tolerance et al, while it’s still unclear whether there are specific differences between L.rhamnosus MS27 and other L.rhamnosus species (such as LGG as mentioned in the introduction and also broadly studies in IBD)? The comparison study between L.rhamnosus MS27 and other Lacticaseibacillus species at least LGG should be conducted to further understand the specific characteristics of MS27 as shown in Figure 1D.
- Figure 1E-G are not convincing taking consideration of the replicates in the figures. The pH, OD value, and CFU numbers are pretty vary in different independent experiments.
- The concentration of DSS used in this study is confusing and should be further confirmed – 3% DSS in both main text and legend of Figure 2, while 5% DSS is used according to Figure 2A.
- The colonization of L.rhamnosus MS27 in different intestinal locations including both small intestinal sections and colon is not clear. As 16s rRNA was performed in this study, can MS27 be detectable in the 16s rRNA dataset?
- It not convincing to indicate that MS27 have positive effects on immune regulation in this study. At least, immune populations which are responsible for chemokines or cytokines production should be investigated via flow cytometry.
- Moreover, it’s recommended to do a correlation analysis between microbial species with chemokines or cytokines-producing immune cells to further understand whether MS27 could modulate immune function via gut microbiota.
- How MS27 regulates DSS-induced inflammation is not understood. Can MS27, as a microbial strain, produce metabolites such as short-chain fatty acids or bile acids? If MS27 itself could not produce sufficient metabolites to play an anti-inflammatory role, whether MS27 could trigger the production of microbiota-derived metabolites in vivo?
Author Response
Dear editor and reviewers,
Thank you very much for taking the time to review this manuscript. I will point-by-point response to comments and suggestions for every reviewer. Here are the detailed responses below and the corresponding corrections highlighted in track changes in the re-submitted files.
For reviewer 1
Comments 1: In the context of introduction (line 50-52), the authors state that conventional therapies such as 5-aminosalicylic acid, glucocorticoids, and et al result in side effects in UC therapy. As such, it’s highly recommended to explain – a. what kind of side effects are basically induced by the above therapies; b. why those therapies cause side effects in UC patients, whether the effects of the above are relevant to gut microbiota. At least, there should be references cited here.
Response 1: Thank you for your constructive advices, which remind me to state my point more clearly. Following it, I supplemented new contends in line 52 and marked the new sentences with red colors, the content of the new paragraph in line 52-68 of Page 2, is “During the various treatment processes for UC, conventional treatments, including 5-aminosalicylic acid (5-ASA), glucocorticoids, and biological agents, have failed or come with significant side effects on a substantial proportion of patients with UC. 5-ASA intolerance is associated with a risk of adverse clinical outcomes and intestinal microbial ecology dysbiosis in patients with UC, its common clinical adverse events include diarrhea, fever and rash [3], and more serious events such as renal toxicity, liver dysfunction, pancreatitis, pericarditis, pneumonia, severe skin reactions, etc. [4]. Glucocorticoid could induce osteoporosis (GIOP) for patients [5], and biological agents, such as monoclonal antibodies infliximab, which working as antitumor necrosis factor (TNF) agents may increase the risk of infection in patients and the risk of certain malignant tumors (such as lymphoma), and response to infliximab therapy is highly variable among individuals, too. Formation of antidrug antibodies (ADAs) for failure of anti-TNF therapy is another thorny issue, especially in children, as the available alter-native treatment options are limited [6]. However, in patients with UC who have undergone ileal pouch-anal anastomosis and experience intermittent symptoms of pouchitis, probiotics is a recommended choice to prevent the recurrence [7]. Whether probiotics can play a role in the prevention of UC is a meaningful exploration option.” And we have added five new references marking as [3], [4], [5], [6], [7] in Page 23, line 844-853.
The additional references are as follows:
[3] Shinta Mizuno, Keiko Ono, et al. 5-Aminosalicylic acid intolerance is associated with a risk of adverse clinical outcomes and dysbiosis in patients with ulcerative colitis. Intest Res. 2020, 18(1):69–78.
[4] Yohei Mikami, Junya Tsunoda, et al. Significance of 5-Aminosalicylic Acid intolerance in the clinical management of ulcerative. Colitis. Digestion. 2022,104(1):58–65.
[5] Mary Beth Humphrey, Linda Russel, et al. 2022 American college of rheumatology guideline for the prevention and treatment of glucocorticoid-induced osteoporosis. Arthritis Rheumatol. 2023, 75(12):2088-2102.
[6] Taku Kobayashi, Satoshi Motoya, Shiro Nakamura, et al. Discontinuation of infliximab in patients with ulcerative colitis in remission (HAYABUSA): a multicentre, open-label, randomised controlled trial. Lancet Gastroenterol Hepatol. 2021, 6(6):429-437.
[7] Edward L Barnes, Manasi Agrawal, et al. AGA clinical practice guideline on the management of pouchitis and inflammatory pouch disorders. Gastroenterology. 2024,166(1):59-85.
Comments 2: Figure 1 characterized the feature of L.rhamnosus MS27 via colony, pH, growth curve, acid tolerance et al, while it’s still unclear whether there are specific differences between L.rhamnosus MS27 and other L.rhamnosus species (such as LGG as mentioned in the introduction and also broadly studies in IBD)? The comparison study between L.rhamnosus MS27 and other Lacticaseibacillus species at least LGG should be conducted to further understand the specific characteristics of MS27 as shown in Figure 1D.
Response 2: Thank you for your thoughtful review. We acknowledge that it was an oversight on our part not to have considered this aspect previously. Your suggestion is highly valuable in characterizing the traits of MS27. Following your suggestion, we first incorporated the core characteristic gene (p 40) sequences of LGG (Lactobacillus rhamnosus GG whole genome sequence, strain GG ATCC 53103, GenBank: FM179322.1) for construct a new phylogenetic tree using Mega software to replace the original one in Figure 1 D. The results confirmed that MS27 exhibits functional differences from LGG in core gene and also differs from other Lactobacillus rhamnosus strains at the gene sequence level. The revised Figure 1 D is as follows (Page 4, line 116):
Secondly, we further conducted a biochemical analysis to compare the carbohydrates metabolic characteristics of several Lactobacillus rhamnosus strains. And reversed the Table 1 (Page 5, line 134). The results confirmed that the ability of MS27 to utilize lactose is distinct among the others, which should be an advantage for high-lactose consumption as an inducer of diarrhea. The revised Table 1 is as follows:
Table 1. Fermented carbohydrates comparison of L. rhamnosus MS27 and other Lactobacillus rhamnosus strains
|
Items |
L.rhamnosus MS27 |
L.rhamnosus LGG [15] |
L.rhamnosus NBRC3425 [16] |
L.rhamnosus JCM1136 [17] |
|
Sucrose |
+ |
+ |
+ |
+ |
|
Lactose |
+ |
- |
- |
- |
|
Maltose |
+ |
+ |
+ |
+ |
|
Inulin |
+ |
+ |
/ |
/ |
|
Synanthrin |
+ |
+ |
+ |
+ |
|
Raffinose |
- |
- |
/ |
/ |
|
Esculin |
+ |
+ |
/ |
+ |
|
Salicin |
+ |
/ |
/ |
/ |
|
Mannitol |
+ |
+ |
+ |
+ |
|
Sorbitol |
+ |
/ |
/ |
+ |
Note: +, means the positive effect; - means the negative effect; / means not data to support it.
And we have added five new references marking as [15], [16], [17] (Page 24, line 868-873). The additional references are as follows:
- Julia B Ewaschuk, Jonathan M Naylor, et al. Lactobacillus rhamnosus strain GG is a potential probiotic for calves. Can J Vet Res J. 2004, 68(4):249–253.
- Xuedi Huang, Fang Ai, et al. A rapid screening method of candidate probiotics for inflammatory bowel diseases and the anti-inflammatory effect of the selected strain bacillus smithii XY1. Front Microbiol. 2021, 17:12:760385.
- Jeffrey E Christensen, Cory E Reynolds, et al. Rapid molecular diagnosis of Lactobacillus Bacteremia by terminal restriction fragment length polymorphism analysis of the 16S rRNA gene. Clin Med Res. 2004, 2(1):37–45.
Comments 3: Figure 1E-G are not convincing taking consideration of the replicates in the figures. The pH, OD value, and CFU numbers are pretty vary in different independent experiments.
Response 3: Thank you for your review. These figures exhibit certain imperfections, so we redid these three experiments and replaced the relevant pictures in Figure 1 E, F, G (Page 4, line 116).
Figure 1. Identification of L. rhamnosus MS27. (A) Colony morphology of L. rhamnosus MS27. bar =20 mm. (B) Gram staining of L. rhamnosus MS27, bar =10 μm. (C) Scanning electron microscope of L. rhamnosus MS27 morphology, enlarge 15000 times, 3 μm. (D) The phylogenetic tree of L. rhamnosus MS27. (E) Acid tolerance of L. rhamnosus MS27. (F) The growth curve of L. rhamnosus MS27. (G) Bacterial count curve of L. rhamnosus MS27.
Comments 4: The concentration of DSS used in this study is confusing and should be further confirmed – 3% DSS in both main text and legend of Figure 2, while 5% DSS is used according to Figure 2A.
Response 4: Thank you for your review. In our trial, 3% DSS was used as the treatment method. We sincerely apologize for the mistake in the original text. It has been corrected, and the figure has been replaced with an updated version in Figure 2 A (Page 6, line 178).
Comments 5: The colonization of L.rhamnosus MS27 in different intestinal locations including both small intestinal sections and colon is not clear. As 16s rRNA was performed in this study, can MS27 be detectable in the 16s rRNA dataset?
Response 5: Thank you for pointing this out. In the 16s rRNA test, the content of Lactobacillus rhamnosus between the DSS group and the LR + DSS group shows a significant difference exactly. MS27 is a novel strain recently isolated by our research team. Prior to its submission, the genetic sequence of MS27 was not present in public gene databases. However, sequence analysis using the NCBI BLAST program confirmed that MS27 belongs to the species Lactobacillus rhamnosus. Therefore, the primary marker to be detected in the 16S rRNA assay should be Lactobacillus rhamnosus, rather than the strain-specific designation Lactobacillus rhamnosus MS27. Figure 3G shows a significant difference in the abundance of Lactobacillus rhamnosus between the two groups; however, this difference is not readily apparent due to its relatively low relative abundance. To better illustrate this comparison, Figure 3 D has been revised to focus the single-species comparison of L. rhamnosus between the DSS group and the LR + DSS group. The revised Figure 3 D (Page 8, line 213) and figure legends (Page 8, line 214-235) are as follows:
Figure 3. (A) PCoA analysis to explore the differences in biodiversity of community compositions among different groups. A two-dimensional visual scatter plot displays the similarity and difference degree of the different communities among groups. The distance between samples reflects the degree of clustering and dispersion of the sample communities. This analysis result is statistically analyzed using ANOSIM/adonis/PERMAVONA, presenting the significant differences in the community of the control group and the treatment group. The scale of the X-axis and Y-axis is relative distance, which has no practical meaning; points of different colors or shapes represent samples of different group. The closer the two sample points are, the more similar the species composition of the two samples is. (B) Beta diversity analysis for inter-group differences between samples. Different distance methods calculate the distance matrices, which are the distance indices of Beta diversity. Statistical tests are conducted on the distance differences between different groups to reflect the degree of sample dispersion within the groups. And this result is consistent with the inter-group variation form of the Shannon index in alpha diversity. (C) NST analysis to uncover the underlying mechanisms of community formation, assess and predict the responses of communities to environmental changes. Based on the algorithm of Jaccard distance, as well as the FDR multiple testing correction method. The horizontal axis represents the sample groups, the vertical axis represents the NST values, and the dotted line in the figure is the threshold for the division between certainty and randomness, indicating the significance of the differences between groups. The box plot shows the maximum value, upper quartile value, median value, lower quartile value, and minimum value. (D) Single-species comparisons of L.rhamnosus between the DSS group and the LR + DSS group. FDR multiple testing correction was adopted; the confidence level of bootstrap was calculated with 0.95.
Comments 6: It not convincing to indicate that MS27 have positive effects on immune regulation in this study. At least, immune populations which are responsible for chemokines or cytokines production should be investigated via flow cytometry.
Response 6: Thanks for your review. This is a very constructive advice and we agree with your comment consistently. So, the animal experiments were redone to obtain fresh spleen samples for flow cytometry detection. This is also the reason why we have delayed returning the revised draft for so long. The content of new paragraph is: (Page 11, line 323-339; Page 12, 340-349)
“2.7 Effects on immune phenotypes of L. rhamnosus MS27
Whether L. rhamnosus MS27 could influence immune phenotypes after RNA-seq identified differentiated expressed genes and STING pathway analysis revealed their enrichment within the immune axis is becoming the next pressing question. Thus, flow cytometry was performed to further explore its immunological phenotypes which predicted by bioinformatics. The results indicated that the population of splenic CD45⁺ CD11b⁺ MHC II⁺ cells increased significantly (P=0.0382) compared with the control group following with L. rhamnosus MS27 treatment (Figure 5A-D) suggesting a proba-ble enhanced antigen-presenting capacity derived from the myeloid lineage. However, the population of splenic CD45+ CD11b+ F4/80- MHC II+ (Figure 5E); CD45+ CD11b+ F4/80+ CD68+ (Figure 5F) and CD45+ CD11b+ F4/80+ CD68+ CD64+ (Figure 5G) were no significant differences between three groups.
To further analyze the systemic inflammatory phenotype, the concentration of serum C-reactive protein (CRP) was also detected. The results shown that the inflammatory stimulation of DSS significantly increased the CRP concentration (P=0.0006), but its concentration significantly decreased (P=0.0428) following the prevention treatment with L. rhamnosus MS27 (Figure 5H)”.
Figure 5: (A) The quantity percentage of CD45+, CD11b+ and MHC II+ cells in control group, n=4. (B, C) the quantity percentage of CD45+, CD11b+ and MHC II+ cells in DSS and LR+DSS treatment groups, n=4 in each treatment. All samples were processed using the same gating strategy. The quantity percentage of non-viable cells is averagely 8.357±0.646(%). (D) The quantity percentage of CD45+ CD11b+ MHC II+ cells in three groups shown with violin plot, P < 0.05 means a significant difference. n = 4. (E) The quantity percentage of CD45+ CD11b+ F4/80- MHC II+ cells in three groups, n = 4. (F) The quantity percentage of CD45+ CD11b+ F4/80+ CD68+ cells in three groups, n = 4. (G) The quantity percentage of CD45+ CD11b+ F4/80+ CD68+ CD64+ cells in three groups, n = 4. (H) Serum CRP concentration in three groups, P < 0.05 means a significant difference. n = 8.
Comments 7: Moreover, it’s recommended to do a correlation analysis between microbial species with chemokines or cytokines-producing immune cells to further understand whether MS27 could modulate immune function via gut microbiota.
Response 7: Thanks for your suggestion. After obtaining the quantity of immune cells through flow cytometry experiments, we utilized the data analysis platform of Majorbio Bio-Pharm Technology Co., Ltd. (Shanghai, China) to analyze the correlation between immune cells and microbial species. The content of new paragraph is: (Page 12, line 350-366; Page 13, 367-381)
“2.8 Correlation analysis between the microbial species and immune cells via L. rhamnosus MS27 treatment
In order to explore whether MS27 could modulate immune function via gut microbiota, the correlation analysis between the microbial species and the immune cells was detected. The results shown that Lachnospiraceae_NK4A136_group, Colidextribacter are significantly positive related to CD45+ CD11b+ F4/80+ CD68+ CD64+ (P < 0.05) and CD45+ CD11b+ F4/80+ CD68+ (P < 0.05) separately, while Clostridium_sensu_stricto_1 is signifi-cantly negative related to CD45+ CD11b+ F4/80+ CD68+ CD64+ (P < 0.01), CD45+ CD11b+ F4/80+ CD68+ and CD45+ CD11b+ F4/80+ CD68+ CD206+ (P < 0.05). Turicibacter is significantly negative related to CD45+ CD11b+ F4/80+ CD68+ CD64+ (P < 0.05) (Figure 6A).
Next the Combine with the Network analysis between the microbial species and the immune cells was detected too. The results shown that Clostridium_sensu_stricto_1 and Turicibacter are negative related to CD45+ CD11b+ F4/80+ CD68+ CD64+, and Clos-tridium_sensu_stricto_1 is negative related to CD45+ CD11b+ F4/80+ CD68+ and CD45+ CD11b+ F4/80+ CD68+ CD206+, but norank_f_Oscillospiraceae is positive related to CD45+ CD11b+ F4/80+ CD68+. Besides these, norank_o_Clostridia_UCG-014 is negative related to CD45+ CD11b+ MHC II+, norank_f_Oscillospiraceae is positive related to CD45+ CD11b+ (Figure 6B).”
Figure 6: (A) The spearman correlation heatmap shows the correlation between species and environmental factors (splenic immune cells). The X-axis and Y-axis represent environmental factors and species respectively. The correlation R value and P value are obtained through calculation. The R value is displayed in different colors in the graph, and the P value is marked with * if it less than 0.05. *, 0.01 < P ≤ 0.05; **, 0.001 < P ≤ 0.01; ***, P ≤ 0.001. (B) The two-factor correlation network diagram focuses on analyzing the correlation between microbial species and environmental factors (splenic immune cells), facilitating the understanding of the interaction between them. The top 50 species are selected, and the Spearman rank correlation and other correlation coefficients between species are calculated to reflect the correlation among species, species with P < 0.05 are displayed. The size of the nodes represents the abundance of the species, and different colors represent different species. The color of the lines with red represents positive correlation and green color represents negative correlation. The thickness of the lines represents the magnitude of the correlation coefficient and the thicker line means the higher correlation. The more lines mean the closer connection between the nodes. (C) The volatile fatty acids and fatty acids production of L. rhamnosus MS27. ****, P ≤ 0.0001.
Comments 8: How MS27 regulates DSS-induced inflammation is not understood. Can MS27, as a microbial strain, produce metabolites such as short-chain fatty acids or bile acids? If MS27 itself could not produce sufficient metabolites to play an anti-inflammatory role, whether MS27 could trigger the production of microbiota-derived metabolites in vivo?
Response 8: Thank you very much for your review. The question you raised is very meaningful. In order to explain this question, we detected the volatile fatty acids and fatty acids production of L. rhamnosus MS27 and add Table 3, Table 4 to illustrate the results. The content of new paragraph is: (Page 13, line 382-392)
“2.9 The volatile fatty acids (VFAs) and fatty acids (FAs) production of L. rhamnosus MS27
In order to explore if the metabolites produced by L. rhamnosus MS27 also playing an important role, short-chain fatty acids (SCFAs) and FAs were detected, too. The re-sults shown that acetic acid is the main metabolites (Figure 6C) with the highest con-centration (Table 3).
Table 3. The volatile fatty acids production of L. rhamnosus MS27
|
Acetic acid (μg/mL) |
Propionic acid (μg/mL) |
Isobutyric acid (μg/mL) |
Butyric acid (μg/mL) |
Isovaleric acid (μg/mL) |
Valeric acid (μg/mL) |
|
3944.11±2.13 |
31.55±0.49 |
7.95±0.01 |
69.58±0.20 |
14.82±0.19 |
11.11±0.06 |
Note: All numerical value denotes with mean ± SEM; n = 3.
Meanwhile, the monounsaturated fatty acids oleic acid is the main fatty acid than others (Table 4).
Table 4. The fatty acids production of L. rhamnosus MS27
|
Oleic acid (μg/mL) |
Stearic acid(μg/mL) |
Palmitic acid(μg/mL) |
Octanoic acid (μg/mL) |
|
230.06±0.10 |
50.07±0.05 |
70.09±0.04 |
40.12±0.05 |
Note: All numerical value denotes with mean ± SEM; n = 3.”
And the contents of discussion parts have been revised too, in Page 16-17, line 537-580.

Reviewer 2 Report
Comments and Suggestions for Authors
This manuscript estimates the probiotic potential of Lacticaseibacillus rhamnosus MS27 for treating ulcerative colitis through gut microbiota modulation. The research uses a comprehensive multi-omics approach combining bacterial characterization, animal experiments, microbiome analysis, and transcriptomics. The work aligns well with other modern findings.
I will allow myself to express my very subjective opinion that it seems to by a typical example of “yet another probiotic discovered” publication. While the benefits of such publications are obvious for the future meta-analyses (and I admit I've published such papers myself), their scientific novelty typically lies solely in the new strain used in the work. Whether this is sufficient for publication in IJMS is up to the editor, I believe. If so, please disregard my comment, as I see no significant issues regarding the scientific accuracy or adequacy of the paper's design. There are a lot of typos and mismatches, though, so please check the text again.
Here are some minor corrections I have:
Line 36: “results also supported that positively regulates immune responses”, - seems like “it” or other subject is missing
Line 44-45: "Its pathogenesis associated with an imbalance gut microbiota" - imbalanced
"which non-pathogenic enteric bacteria plays a primary role on" - in which
Line 54-55: "marked by a decrease in characterized by a decrease in" – repeat
Line 73: " character of L. rhamnosus" – characteristics
Line 95: "MS27 is a growth-producing acid bacterium" - "MS27 is an acid-producing bacterium "
Line 310: “Evidence from several strains indicates a critical role in” – their critical role
Line 346: “ L. rhamnosus MS27 significantly improved the intestinal microbial flora is regarded as an important way for it to exert beneficial effects” – “which is regarded”, maybe?
Line 495: “The drug of the strains was evaluated” – the drug resistance
Regarding transcriptional changes, I’d suggest to discuss potential pathways, through which these specific gene expression decrease might contribute to the anti-inflammatory effects.
Also, given the apparent absence of pili structures, what supports MS27's ability to effectively colonize surfaces and form biofilms, which depends partially on pili in other L. rhamnosus strains?
In general, if the authors highlight somewhere in the introduction and/or conclusion what is new about their work, other than the new strain, this will greatly enhance the manuscript.
The conclusion itself is rather short and would benefit from adding several main points summarizing the study’s findings.
Author Response
Dear reviewers,
Thank you very much for taking the time to review this manuscript. I will point-by-point response to comments and suggestions for every reviewer. Here are the detailed responses below and the corresponding corrections highlighted in track changes in the re-submitted files.
For reviewer 2
This manuscript estimates the probiotic potential of Lacticaseibacillus rhamnosus MS27 for treating ulcerative colitis through gut microbiota modulation. The research uses a comprehensive multi-omics approach combining bacterial characterization, animal experiments, microbiome analysis, and transcriptomics. The work aligns well with other modern findings.
I will allow myself to express my very subjective opinion that it seems to by a typical example of “yet another probiotic discovered” publication. While the benefits of such publications are obvious for the future meta-analyses (and I admit I've published such papers myself), their scientific novelty typically lies solely in the new strain used in the work. Whether this is sufficient for publication in IJMS is up to the editor, I believe. If so, please disregard my comment, as I see no significant issues regarding the scientific accuracy or adequacy of the paper's design. There are a lot of typos and mismatches, though, so please check the text again. Here are some minor corrections I have:
Comments 1: Line 36: “results also supported that positively regulates immune responses”, - seems like “it” or another subject is missing
Response 1: Thanks for your suggestion. I have revised the whole sentence, the new sentence is “L. rhamnosus MS27 demonstrated a significant ability to alleviate inflammatory phenotypes; research findings further indicate its potential to positively modulate immune responses, a mechanism that may underlie its beneficial effects on inflammation” in Page 1, line 35-39.
Comments 2: Line 44-45: "Its pathogenesis associated with an imbalance gut microbiota" - imbalanced
Response 2: Thank you for pointing it out. I have revised “an imbalance” to “a dysbiosis of” in Page 2, line 45.
Comments 3: "which non-pathogenic enteric bacteria plays a primary role on" - in which
Response 3: Thanks, I have added a word “in” before “which” to make this sentence more accurate. This revision is in Page 2, line 46.
Comments 4: Line 54-55: "marked by a decrease in characterized by a decrease in" – repeat
Response 4: I am so sorry to make this mistake. I have deleted the repeat phrase “by a decrease”, so the new sentence is “marked in characterized by a decrease in beneficial bacteria and an increase in pathogenic bacteria” in Page 2, line 69-70.
Comments 5: Line 73: " character of L. rhamnosus" – characteristics
Response 5: Thank you for your careful review, I have revised it as “characteristics of L. rhamnosus" in Page 2, line 85.
Comments 6: Line 95: "MS27 is a growth-producing acid bacterium" - "MS27 is an acid-producing bacterium "
Response 6: Thanks for your careful review to help reducing the ambiguity of the sentence. I have revised this sentence as “MS27 is an acid-producing bacterium” in Page 3, line 106.
Comments 7: Line 310: “Evidence from several strains indicates a critical role in” – their critical role
Response 7: Thank you point this. I have revised the word “a” to “their” in Page 13, line 392.
Comments 8: Line 346: “L. rhamnosus MS27 significantly improved the intestinal microbial flora is regarded as an important way for it to exert beneficial effects” – “which is regarded”, maybe?
Response 8: I agree with your proposed modifications. I have revised this sentence as “L. rhamnosus MS27 significantly improved the intestinal microbial flora, which is regarded as” in Page 13, line 428-429.
Comments 9: Line 495: “The drug of the strains was evaluated” – the drug resistance
Response 9: I fully agree with your proposed modifications. I have revised this sentence as “The drug resistance of the strains was evaluated” in Page 17, line 611.
Comments 10: Regarding transcriptional changes, I’d suggest to discuss potential pathways, through which these specific gene expression decrease might contribute to the anti-inflammatory effects.
Response 10: Thanks for your suggestion. We discuss a potential pathway of gene Igkv16-104 contribute to the anti-inflammatory effects in Page 14-15, line 491-501. The new paragraph is “ Under the condition of inflammatory bowel disease, the expression of Igkv16-104 is related to immune regulation and it should connect with IL6 expression [35]. In inflammatory bowel diseases (IBD) and other inflammatory disorders, the level of IL-6 usually increases. It should be that intestinal inflammation stimulates macrophag-es/Th17 to secrete IL-6, which then activates the JAK-STAT3 pathway of B cell, thereby upregulating the Igκ enhancer expression (like Igkv16-104) and promoting antibody secretion, thereby exacerbating inflammation or tissue damage”. In Page 14-15, line 486-493.
Comments 11: Also, given the apparent absence of pili structures, what supports MS27's ability to effectively colonize surfaces and form biofilms, which depends partially on pili in other L. rhamnosus strains?
Response 11: Your observation is careful. The apparent absence of pilus structures likely affects the colonization capacity of MS27 in the intestinal tract, but like the other probiotics, such as Lactobacillus plantarum, which have no pili too, MS27 should keeping its beneficial effects via their ability of acid production and space occupying, it may have the function similarly with transient probiotics, which require frequent administration to exert beneficial effects. In this study, we did not explore the effect of its colonization, but we achieved continuous supply through intragastric administration every other day from 0-14 days. And the results also confirmed its role in improving immune homeostasis within the host system. Future work will focus on further elucidating its mechanism of action to support broader therapeutic applications.
Comments 12: In general, if the authors highlight somewhere in the introduction and/or conclusion what is new about their work, other than the new strain, this will greatly enhance the manuscript.
Response 12: Thanks for your beneficial suggestion. Following your suggestion, we changed the content in introduction, the new sentence is “In this study, L. rhamnosus MS27 demonstrated its role in alleviating inflammatory phenotypes surprisingly. Then we will explore the comprehensive characteristics of L. rhamnosus MS27 and its effectiveness of the action from the perspective of microbiomics and untargeted transcriptomics. Through multiomics association analysis and microbiota-immune axis elucidation, new research targets will be established for advancing more depth mechanism analysis on L. rhamnosus MS27.”, Page 2, line 84-89.
And we also revised the conclusion part, too. The new sentence is “As a novel strain, L. rhamnosus MS27 exhibits unique characteristics in lactic acid utilization and acetic acid, oleic acid production. Moreover, we further did an exploratory analysis on the microbiota–immune axis elucidation, highlighted that MS27 improves the abundance of Turicibacter, and demonstrate its negative correlation with pro-inflammatory macrophage subsets (CD45⁺CD11b⁺F4/80⁺CD68⁺CD64⁺), suggesting a novel microbiota–immune pathway for systemic immune homeostasis. These characteristics of L. rhamnosus MS27 may collaboratively contribute to the alleviation of intestinal inflammation.”, Page 21, line 801-807.
Comments 13: The conclusion itself is rather short and would benefit from adding several main points summarizing the study’s findings.
Response 13: Your suggestion is very valuable. Following your suggestion, we further explored the unique metabolic features of MS27, its unique ability to utilize lactose (Page 4, line 121-128) and its characteristic production of high levels of acetic acid and oleic acid. These metabolic traits have not been previously reported in L. rhamnosus strains and may contribute to the anti-inflammatory effects of L. rhamnosus MS27 (Page 12, line 376-386). Moreover, we further did an exploratory analysis on the microbiota–immune axis elucidation, highlighted that MS27 improves the abundance of Turicibacter, and demonstrate its negative correlation with pro-inflammatory macrophage subsets (CD45⁺CD11b⁺F4/80⁺CD68⁺CD64⁺), suggesting a novel microbiota–immune pathway for systemic immune homeostasis (Page 10-12, line 317-375).
And we also revised the conclusion part, too. The new sentence is “As a novel strain, L. rhamnosus MS27 exhibits unique characteristics in lactic acid utilization and acetic acid, oleic acid production. Moreover, we further did an exploratory analysis on the microbiota–immune axis elucidation, highlighted that MS27 improves the abundance of Turicibacter, and demonstrate its negative correlation with pro-inflammatory macrophage subsets (CD45⁺CD11b⁺F4/80⁺CD68⁺CD64⁺), suggesting a novel microbiota–immune pathway for systemic immune homeostasis. These characteristics of L. rhamnosus MS27 may collaboratively contribute to the alleviation of intestinal inflammation.”, Page 21, line 801-807.

Round 2
Reviewer 1 Report
Comments and Suggestions for Authors
The authors addressed all my concerns and did a good job. I would recommend accept after minor editing of format which I didn’t mention in my initial reviewing. Please carefully improve the format and check potential typos through the whole manuscript on the author side. No reviewer comments are needed further.
Author Response
Dear reviewers,
Thank you very much for taking the time to review this manuscript. I will point-by-point response to comments and suggestions for every reviewer. Here are the detailed responses below and the corresponding corrections highlighted in track changes in the re-submitted files.
For reviewer 1
Comments 1: In the context of introduction (line 50-52), the authors state that conventional therapies such as 5-aminosalicylic acid, glucocorticoids, and et al result in side effects in UC therapy. As such, it’s highly recommended to explain – a. what kind of side effects are basically induced by the above therapies; b. why those therapies cause side effects in UC patients, whether the effects of the above are relevant to gut microbiota. At least, there should be references cited here.
Response 1: Thank you for your constructive advices, which remind me to state my point more clearly. Following it, I supplemented new contends in line 52 and marked the new sentences with red colors, the content of the new paragraph in line 50-66 of Page 2, is “During the various treatment processes for UC, conventional treatments, including 5-aminosalicylic acid (5-ASA), glucocorticoids, and biological agents, have failed or come with significant side effects on a substantial proportion of patients with UC. 5-ASA intolerance is associated with a risk of adverse clinical outcomes and intestinal microbial ecology dysbiosis in patients with UC, its common clinical adverse events include diarrhea, fever and rash [3], and more serious events such as renal toxicity, liver dysfunction, pancreatitis, pericarditis, pneumonia, severe skin reactions, etc. [4]. Glucocorticoid could induce osteoporosis (GIOP) for patients [5], and biological agents, such as monoclonal antibodies infliximab, which working as antitumor necrosis factor (TNF) agents may increase the risk of infection in patients and the risk of certain malignant tumors (such as lymphoma), and response to infliximab therapy is highly variable among individuals, too. Formation of antidrug antibodies (ADAs) for failure of anti-TNF therapy is another thorny issue, especially in children, as the available alter-native treatment options are limited [6]. However, in patients with UC who have undergone ileal pouch-anal anastomosis and experience intermittent symptoms of pouchitis, probiotics is a recommended choice to prevent the recurrence [7]. Whether probiotics can play a role in the prevention of UC is a meaningful exploration option.” And we have added five new references marking as [3], [4], [5], [6], [7] in Page 22, line 836-845.
The additional references are as follows:
[3] Shinta Mizuno, Keiko Ono, et al. 5-Aminosalicylic acid intolerance is associated with a risk of adverse clinical outcomes and dysbiosis in patients with ulcerative colitis. Intest Res. 2020, 18(1):69–78.
[4] Yohei Mikami, Junya Tsunoda, et al. Significance of 5-Aminosalicylic Acid intolerance in the clinical management of ulcerative. Colitis. Digestion. 2022,104(1):58–65.
[5] Mary Beth Humphrey, Linda Russel, et al. 2022 American college of rheumatology guideline for the prevention and treatment of glucocorticoid-induced osteoporosis. Arthritis Rheumatol. 2023, 75(12):2088-2102.
[6] Taku Kobayashi, Satoshi Motoya, Shiro Nakamura, et al. Discontinuation of infliximab in patients with ulcerative colitis in remission (HAYABUSA): a multicentre, open-label, randomised controlled trial. Lancet Gastroenterol Hepatol. 2021, 6(6):429-437.
[7] Edward L Barnes, Manasi Agrawal, et al. AGA clinical practice guideline on the management of pouchitis and inflammatory pouch disorders. Gastroenterology. 2024,166(1):59-85.
Comments 2: Figure 1 characterized the feature of L.rhamnosus MS27 via colony, pH, growth curve, acid tolerance et al, while it’s still unclear whether there are specific differences between L.rhamnosus MS27 and other L.rhamnosus species (such as LGG as mentioned in the introduction and also broadly studies in IBD)? The comparison study between L.rhamnosus MS27 and other Lacticaseibacillus species at least LGG should be conducted to further understand the specific characteristics of MS27 as shown in Figure 1D.
Response 2: Thank you for your thoughtful review. We acknowledge that it was an oversight on our part not to have considered this aspect previously. Your suggestion is highly valuable in characterizing the traits of MS27. Following your suggestion, we first incorporated the core characteristic gene (p 40) sequences of LGG (Lactobacillus rhamnosus GG whole genome sequence, strain GG ATCC 53103, GenBank: FM179322.1) for construct a new phylogenetic tree using Mega software to replace the original one in Figure 1 D. The results confirmed that MS27 exhibits functional differences from LGG in core gene and also differs from other Lactobacillus rhamnosus strains at the gene sequence level. The revised Figure 1 D is as follows (Page 3, line 110):
Secondly, we further conducted a biochemical analysis to compare the carbohydrates metabolic characteristics of several Lactobacillus rhamnosus strains. And reversed the Table 1 (Page 4, line 126). The results confirmed that the ability of MS27 to utilize lactose is distinct among the others, which should be an advantage for high-lactose consumption as an inducer of diarrhea. The revised Table 1 is as follows:
Table 1. Fermented carbohydrates comparison of L. rhamnosus MS27 and other Lactobacillus rhamnosus strains
|
Items |
L.rhamnosus MS27 |
L.rhamnosus LGG [15] |
L.rhamnosus NBRC3425 [16] |
L.rhamnosus JCM1136 [17] |
|
Sucrose |
+ |
+ |
+ |
+ |
|
Lactose |
+ |
- |
- |
- |
|
Maltose |
+ |
+ |
+ |
+ |
|
Inulin |
+ |
+ |
/ |
/ |
|
Synanthrin |
+ |
+ |
+ |
+ |
|
Raffinose |
- |
- |
/ |
/ |
|
Esculin |
+ |
+ |
/ |
+ |
|
Salicin |
+ |
/ |
/ |
/ |
|
Mannitol |
+ |
+ |
+ |
+ |
|
Sorbitol |
+ |
/ |
/ |
+ |
Note: +, means the positive effect; - means the negative effect; / means not data to support it.
And we have added five new references marking as [15], [16], [17] (Page 23, line 860-865). The additional references are as follows:
- Julia B Ewaschuk, Jonathan M Naylor, et al. Lactobacillus rhamnosus strain GG is a potential probiotic for calves. Can J Vet Res J. 2004, 68(4):249–253.
- Xuedi Huang, Fang Ai, et al. A rapid screening method of candidate probiotics for inflammatory bowel diseases and the anti-inflammatory effect of the selected strain bacillus smithii XY1. Front Microbiol. 2021, 17:12:760385.
- Jeffrey E Christensen, Cory E Reynolds, et al. Rapid molecular diagnosis of Lactobacillus Bacteremia by terminal restriction fragment length polymorphism analysis of the 16S rRNA gene. Clin Med Res. 2004, 2(1):37–45.
Comments 3: Figure 1E-G are not convincing taking consideration of the replicates in the figures. The pH, OD value, and CFU numbers are pretty vary in different independent experiments.
Response 3: Thank you for your review. These figures exhibit certain imperfections, so we redid these three experiments and replaced the relevant pictures in Figure 1 E, F, G (Page 3, line 110).
Figure 1. Identification of L. rhamnosus MS27. (A) Colony morphology of L. rhamnosus MS27. bar =20 mm. (B) Gram staining of L. rhamnosus MS27, bar =10 μm. (C) Scanning electron microscope of L. rhamnosus MS27 morphology, enlarge 15000 times, 3 μm. (D) The phylogenetic tree of L. rhamnosus MS27. (E) Acid tolerance of L. rhamnosus MS27. (F) The growth curve of L. rhamnosus MS27. (G) Bacterial count curve of L. rhamnosus MS27.
Comments 4: The concentration of DSS used in this study is confusing and should be further confirmed – 3% DSS in both main text and legend of Figure 2, while 5% DSS is used according to Figure 2A.
Response 4: Thank you for your review. In our trial, 3% DSS was used as the treatment method. We sincerely apologize for the mistake in the original text. It has been corrected, and the figure has been replaced with an updated version in Figure 2 A (Page 5, line 172).
Comments 5: The colonization of L.rhamnosus MS27 in different intestinal locations including both small intestinal sections and colon is not clear. As 16s rRNA was performed in this study, can MS27 be detectable in the 16s rRNA dataset?
Response 5: Thank you for pointing this out. In the 16s rRNA test, the content of Lactobacillus rhamnosus between the DSS group and the LR + DSS group shows a significant difference exactly. MS27 is a novel strain recently isolated by our research team. Prior to its submission, the genetic sequence of MS27 was not present in public gene databases. However, sequence analysis using the NCBI BLAST program confirmed that MS27 belongs to the species Lactobacillus rhamnosus. Therefore, the primary marker to be detected in the 16S rRNA assay should be Lactobacillus rhamnosus, rather than the strain-specific designation Lactobacillus rhamnosus MS27. Figure 3G shows a significant difference in the abundance of Lactobacillus rhamnosus between the two groups; however, this difference is not readily apparent due to its relatively low relative abundance. To better illustrate this comparison, Figure 3 D has been revised to focus the single-species comparison of L. rhamnosus between the DSS group and the LR + DSS group. The revised Figure 3 D (Page 6, line 207) and figure legends (Page 7, line 208-229) are as follows:
Figure 3. (A) PCoA analysis to explore the differences in biodiversity of community compositions among different groups. A two-dimensional visual scatter plot displays the similarity and difference degree of the different communities among groups. The distance between samples reflects the degree of clustering and dispersion of the sample communities. This analysis result is statistically analyzed using ANOSIM/adonis/PERMAVONA, presenting the significant differences in the community of the control group and the treatment group. The scale of the X-axis and Y-axis is relative distance, which has no practical meaning; points of different colors or shapes represent samples of different group. The closer the two sample points are, the more similar the species composition of the two samples is. (B) Beta diversity analysis for inter-group differences between samples. Different distance methods calculate the distance matrices, which are the distance indices of Beta diversity. Statistical tests are conducted on the distance differences between different groups to reflect the degree of sample dispersion within the groups. And this result is consistent with the inter-group variation form of the Shannon index in alpha diversity. (C) NST analysis to uncover the underlying mechanisms of community formation, assess and predict the responses of communities to environmental changes. Based on the algorithm of Jaccard distance, as well as the FDR multiple testing correction method. The horizontal axis represents the sample groups, the vertical axis represents the NST values, and the dotted line in the figure is the threshold for the division between certainty and randomness, indicating the significance of the differences between groups. The box plot shows the maximum value, upper quartile value, median value, lower quartile value, and minimum value. (D) Single-species comparisons of L.rhamnosus between the DSS group and the LR + DSS group. FDR multiple testing correction was adopted; the confidence level of bootstrap was calculated with 0.95.
Comments 6: It not convincing to indicate that MS27 have positive effects on immune regulation in this study. At least, immune populations which are responsible for chemokines or cytokines production should be investigated via flow cytometry.
Response 6: Thanks for your review. This is a very constructive advice and we agree with your comment consistently. So, the animal experiments were redone to obtain fresh spleen samples for flow cytometry detection. This is also the reason why we have delayed returning the revised draft for so long. The content of new paragraph is: (Page 10, line 317-340; Page 11, 341-343)
“2.7 Effects on immune phenotypes of L. rhamnosus MS27
Whether L. rhamnosus MS27 could influence immune phenotypes after RNA-seq identified differentiated expressed genes and STING pathway analysis revealed their enrichment within the immune axis is becoming the next pressing question. Thus, flow cytometry was performed to further explore its immunological phenotypes which predicted by bioinformatics. The results indicated that the population of splenic CD45⁺ CD11b⁺ MHC II⁺ cells increased significantly (P=0.0382) compared with the control group following with L. rhamnosus MS27 treatment (Figure 5A-D) suggesting a proba-ble enhanced antigen-presenting capacity derived from the myeloid lineage. However, the population of splenic CD45+ CD11b+ F4/80- MHC II+ (Figure 5E); CD45+ CD11b+ F4/80+ CD68+ (Figure 5F) and CD45+ CD11b+ F4/80+ CD68+ CD64+ (Figure 5G) were no significant differences between three groups.
To further analyze the systemic inflammatory phenotype, the concentration of serum C-reactive protein (CRP) was also detected. The results shown that the inflammatory stimulation of DSS significantly increased the CRP concentration (P=0.0006), but its concentration significantly decreased (P=0.0428) following the prevention treatment with L. rhamnosus MS27 (Figure 5H)”.
Figure 5: (A) The quantity percentage of CD45+, CD11b+ and MHC II+ cells in control group, n=4. (B, C) the quantity percentage of CD45+, CD11b+ and MHC II+ cells in DSS and LR+DSS treatment groups, n=4 in each treatment. All samples were processed using the same gating strategy. The quantity percentage of non-viable cells is averagely 8.357±0.646(%). (D) The quantity percentage of CD45+ CD11b+ MHC II+ cells in three groups shown with violin plot, P < 0.05 means a significant difference. n = 4. (E) The quantity percentage of CD45+ CD11b+ F4/80- MHC II+ cells in three groups, n = 4. (F) The quantity percentage of CD45+ CD11b+ F4/80+ CD68+ cells in three groups, n = 4. (G) The quantity percentage of CD45+ CD11b+ F4/80+ CD68+ CD64+ cells in three groups, n = 4. (H) Serum CRP concentration in three groups, P < 0.05 means a significant difference. n = 8.
Comments 7: Moreover, it’s recommended to do a correlation analysis between microbial species with chemokines or cytokines-producing immune cells to further understand whether MS27 could modulate immune function via gut microbiota.
Response 7: Thanks for your suggestion. After obtaining the quantity of immune cells through flow cytometry experiments, we utilized the data analysis platform of Majorbio Bio-Pharm Technology Co., Ltd. (Shanghai, China) to analyze the correlation between immune cells and microbial species. The content of new paragraph is: (Page 11, line 344-375)
“2.8 Correlation analysis between the microbial species and immune cells via L. rhamnosus MS27 treatment
In order to explore whether MS27 could modulate immune function via gut microbiota, the correlation analysis between the microbial species and the immune cells was detected. The results shown that Lachnospiraceae_NK4A136_group, Colidextribacter are significantly positive related to CD45+ CD11b+ F4/80+ CD68+ CD64+ (P < 0.05) and CD45+ CD11b+ F4/80+ CD68+ (P < 0.05) separately, while Clostridium_sensu_stricto_1 is signifi-cantly negative related to CD45+ CD11b+ F4/80+ CD68+ CD64+ (P < 0.01), CD45+ CD11b+ F4/80+ CD68+ and CD45+ CD11b+ F4/80+ CD68+ CD206+ (P < 0.05). Turicibacter is significantly negative related to CD45+ CD11b+ F4/80+ CD68+ CD64+ (P < 0.05) (Figure 6A).
Next the Combine with the Network analysis between the microbial species and the immune cells was detected too. The results shown that Clostridium_sensu_stricto_1 and Turicibacter are negative related to CD45+ CD11b+ F4/80+ CD68+ CD64+, and Clos-tridium_sensu_stricto_1 is negative related to CD45+ CD11b+ F4/80+ CD68+ and CD45+ CD11b+ F4/80+ CD68+ CD206+, but norank_f_Oscillospiraceae is positive related to CD45+ CD11b+ F4/80+ CD68+. Besides these, norank_o_Clostridia_UCG-014 is negative related to CD45+ CD11b+ MHC II+, norank_f_Oscillospiraceae is positive related to CD45+ CD11b+ (Figure 6B).”
Figure 6: (A) The spearman correlation heatmap shows the correlation between species and environmental factors (splenic immune cells). The X-axis and Y-axis represent environmental factors and species respectively. The correlation R value and P value are obtained through calculation. The R value is displayed in different colors in the graph, and the P value is marked with * if it less than 0.05. *, 0.01 < P ≤ 0.05; **, 0.001 < P ≤ 0.01; ***, P ≤ 0.001. (B) The two-factor correlation network diagram focuses on analyzing the correlation between microbial species and environmental factors (splenic immune cells), facilitating the understanding of the interaction between them. The top 50 species are selected, and the Spearman rank correlation and other correlation coefficients between species are calculated to reflect the correlation among species, species with P < 0.05 are displayed. The size of the nodes represents the abundance of the species, and different colors represent different species. The color of the lines with red represents positive correlation and green color represents negative correlation. The thickness of the lines represents the magnitude of the correlation coefficient and the thicker line means the higher correlation. The more lines mean the closer connection between the nodes. (C) The volatile fatty acids and fatty acids production of L. rhamnosus MS27. ****, P ≤ 0.0001.
Comments 8: How MS27 regulates DSS-induced inflammation is not understood. Can MS27, as a microbial strain, produce metabolites such as short-chain fatty acids or bile acids? If MS27 itself could not produce sufficient metabolites to play an anti-inflammatory role, whether MS27 could trigger the production of microbiota-derived metabolites in vivo?
Response 8: Thank you very much for your review. The question you raised is very meaningful. In order to explain this question, we detected the volatile fatty acids and fatty acids production of L. rhamnosus MS27 and add Table 3, Table 4 to illustrate the results. The content of new paragraph is: (Page 12, line 376-386)
“2.9 The volatile fatty acids (VFAs) and fatty acids (FAs) production of L. rhamnosus MS27
In order to explore if the metabolites produced by L. rhamnosus MS27 also playing an important role, short-chain fatty acids (SCFAs) and FAs were detected, too. The re-sults shown that acetic acid is the main metabolites (Figure 6C) with the highest con-centration (Table 3).
Table 3. The volatile fatty acids production of L. rhamnosus MS27
|
Acetic acid (μg/mL) |
Propionic acid (μg/mL) |
Isobutyric acid (μg/mL) |
Butyric acid (μg/mL) |
Isovaleric acid (μg/mL) |
Valeric acid (μg/mL) |
|
3944.11±2.13 |
31.55±0.49 |
7.95±0.01 |
69.58±0.20 |
14.82±0.19 |
11.11±0.06 |
Note: All numerical value denotes with mean ± SEM; n = 3.
Meanwhile, the monounsaturated fatty acids oleic acid is the main fatty acid than others (Table 4).
Table 4. The fatty acids production of L. rhamnosus MS27
|
Oleic acid (μg/mL) |
Stearic acid(μg/mL) |
Palmitic acid(μg/mL) |
Octanoic acid (μg/mL) |
|
230.06±0.10 |
50.07±0.05 |
70.09±0.04 |
40.12±0.05 |
Note: All numerical value denotes with mean ± SEM; n = 3.”
And the contents of discussion parts have been revised too, in Page 15-16, line 531-570.
